

# Theia Snow collection: high resolution operational snow cover maps from Sentinel-2 and Landsat-8 data

Simon Gascoin[1], Manuel Grizonnet[2], Marine Bouchet[1,2], Germain Salgues[2,3], and Olivier Hagolle[1,2]

[1]CESBIO, Université de Toulouse, CNRS/CNES/IRD/INRA/UPS, Toulouse, France
[2]CNES, Toulouse, France
[3]Magellium, Toulouse, France

**Correspondence:** Simon Gascoin (simon.gascoin@cesbio.cnes.fr)

**Abstract.** The Theia Snow collection routinely provides high resolution maps of the snow cover area from Sentinel-2 and Landsat-8 observations. The collection covers selected areas worldwide including the main mountain regions in Western Europe (e.g. Alps, Pyrenees) and the High Atlas in Morocco. Each product of the Snow collection contains four classes: snow, no-snow, cloud and no-data. We present the algorithm to generate the snow products and provide an evaluation of their accuracy
using in situ snow depth measurements, higher resolution snow maps, and visual control. The results suggest that the snow is accurately detected in the Theia snow collection, and that the snow detection is more accurate than the sen2cor outputs (ESA level 2 product). An issue that should be addressed in a future release is the occurrence of false snow detection in some large clouds. The snow maps are currently produced and freely distributed in average 5 days after the image acquisition as raster and vector files via the Theia portal (DOI:10.24400/329360/F7Q52MNK).

## 1   Introduction

The snow cover is an important driver of many ecological, climatic and hydrological processes in mountain regions and in high latitude areas. In these regions, in situ observations are generally insufficient to characterize the high spatial variability of the snowpack properties. The snow cover was included as one of the 50 "Essential Climate Variables" (ECVs) to be monitored by satellite remote sensing by the Global Observing System for Climate (GCOS) in accordance with the Committee on Earth Observation Satellites (CEOS) agencies. ECVs are intended to support the work of the United Nations Framework Convention
on Climate Change and the Intergovernmental Panel on Climate Change.

The snow cover is not a variable *sensu stricto*, but an object which can be characterized through many variables, including: snow cover area (SCA), fractional area (fSCA), albedo, liquid water content, snow depth and snow water equivalent (Frei et al., 2012). An international survey by the Cryoland consortium[1] shed light on the user requirements for snow services based on
satellite remote sensing. SCA/fSCA products were ranked as the most important products by the respondents Malnes et al. (2015). The survey also indicated the need for operational products provided on a daily basis, with latency time shorter than 12h. High resolution data down to 50 m resolution were sought by road and avalanches authorities (Malnes et al., 2015). The

---

[1]http://cryoland.enveo.at/consortium





respondents requested regional products e.g. at the scale of entire mountain range like the Alps or even Pan-European and preferred the UTM projection (Malnes et al., 2015). At the national level in France there is also a need for operational high resolution snow cover area maps as revealed by the recent roadmap for satellite applications issued by the French Government (*Plan d'applications satellitaires 2018*, Commissariat général au développement durable, 2018). Based on a wide panel of

end-users, this plan selected the monitoring of the snow cover area in French National Parks as a one of the key actions which should be achieved using Earth Observation satellites in the near future (2018-2022).

Operational SCA maps have been generated from satellite observations since the 1960's (Ramsay, 1998). Current SCA and fSCA products are mostly derived from MODIS data (Hall et al., 2002; Sirguey et al., 2008; Painter et al., 2009; Metsämäki et al., 2015), but their spatial resolution (1 km to 250 m) can be too coarse for various applications, especially in mountain

regions where the snow cover properties vary at scales of 10 m to 100 m Blöschl (1999). High resolution (30 m) snow cover maps can be generated from Landsat images but the low temporal revisit of the Landsat mission (16 days) is an important limitation for snow cover monitoring, especially considering that the cloud probability can exceed 50% in temperate mountains (Parajka and Blöschl, 2008; Gascoin et al., 2015). Since 2017 with the launch of Sentinel-2B, the Copernicus Sentinel-2 mission offers the unique opportunity to map the snow cover extent at 20 m resolution with a 5 day revisit time (cloud-permitting)

(Drusch et al., 2012). The combination of Sentinel-2 and Landsat-8 data provides the opportunity for even more frequent observations of the snow cover with a global median average revisit interval of 2.9 days (Li and Roy, 2017).

The principles of snow detection from multispectral optical imagery is well established since the pioneering work of Dozier (1989) with the Landsat Thematic Mapper. Today the challenge for scientists and end-users is rather to cope with the large amount of data that is generated by a mission like Sentinel-2. The generation of a single snow map from a Sentinel-2 level-1C

tiled product involves the download of a 700+ Mo zip file (once uncompressed, the product contains 12 folders and 108 files, including 15 raster files and 13 XML metadata files). Since March 2018, ESA began the operational processing of level 2A (L2A) products. Each ESA L2A product also includes a snow cover mask. However, the size of a single L2A product before unzipping can exceed 1 Go (15 folders, 137 files). In addition, the quality of the ESA L2A snow mask can be improved for two reasons (i) ESA's L2A algorithm is a general purpose algorithm, which was not optimized for snow detection (ii) Because

ESA's L2A processor treats each image independently (i.e. mono-date approach, Louis et al., 2016), the output cloud mask has a lower accuracy than a cloud mask generated by multi-temporal algorithm (Hagolle et al., 2010).

In this article we introduce the Theia Snow collection, a new collection of snow cover maps, which are derived from Sentinel-2 and Landsat-8 imagery at 20 m resolution (30 m respectively) in an operational context. The data are currently being produced and freely distributed via the Theia land data center [2]. The snow retrieval algorithm is based on the Normalized Difference

Snow Index, but also uses a digital elevation model to better constrain the snow detection. We first describe the algorithm (Sect. 2 and its implementation in the let-it-snow processor (LIS, Sect. 2.7). Then, we provide a detailed description of the product characteristics (coverage, period, format, etc., Sect. 3). In Section 4, we present an evaluation of Theia snow products

---

[2]Theia is a French inter-agency organization designed to foster the use of Earth observation for land surface monitoring for academics and public policy actors.



using in situ snow depth measurements, very-high resolution snow maps and visual control. We finally discuss the main limitations and potential applications of the Theia Snow collection.

## 2 Algorithm

### 2.1 Scope

An algorithm was designed to determine the snow presence or absence from Sentinel-2 and Landsat-8 observations with the following requirements:

- It should be scalable, i.e. allow the processing of large areas ($10^4$ km$^2$) with a reasonable computation cost (typically less than 1 hour for a single product).

- It should be robust to seasonal and spatial variability of the snow cover and land surface properties.

- It should maximize the number of pixels that are classified as snow or no-snow.

- It is preferable to falsely classify a pixel as cloud than falsely classify a pixel as snow or no-snow.

### 2.2 Input

The algorithm works with multispectral remote sensing images, which include at least a channel in the visible part of the spectrum and a channel near 1.5 μm (referred to as shortwave-infrared or "SWIR"). It takes as input:

- a L2A product, including:

  - the cloud and cloud shadow mask (referred to as "L2A cloud mask" in the following),

  - the green, red and SWIR bands from the flat surface reflectance product. These images are corrected for atmospheric and terrain slope effects. The slope correction is important in mountain regions since it enables to use the same detection thresholds whatever the sun-slope geometry.

- A digital elevation model (DEM).

### 2.3 Pre-processing

The red and green bands are resampled with the cubic method to a pixel size of 20 m by 20 m to match the resolution of the SWIR band. The DEM is generated from the SRTM seamless DEM (Jarvis et al., 2008) by cubic spline resampling to the same 20 m resolution grid.





## 2.4 Algorithm description

### 2.4.1 Snow detection

The snow detection is based on the Normalized Difference Snow Index (NDSI, Dozier, 1989) and the reflectance in the red band. The NDSI is defined as:

$$\mathrm{NDSI} = \frac{\rho_{\mathrm{green}} - \rho_{\mathrm{SWIR}}}{\rho_{\mathrm{green}} + \rho_{\mathrm{SWIR}}} \tag{1}$$

where $\rho_{\mathrm{green}}$ (resp. $\rho_{\mathrm{SWIR}}$) is the slope-corrected surface reflectance in the green band (resp. SWIR at 1.6 $\mu$m). The NDSI expresses the fact that only snow surfaces are very bright in the visible and very dark in the shortwave infrared. Turbid water surfaces like some lakes or rivers may also have a high NDSI value, hence a additional criterion on the red reflectance is used to avoid false snow detection in these areas. A cloud-free pixel is classified as "snow" if the following condition is true:

$$(\mathrm{NDSI} > n_i) \text{ and } (\rho_{\mathrm{red}} > r_i) \tag{2}$$

where $n_i$ and $r_i$ are two parameters with two possible values each (i.e. $i \in \{1,2\}$) as explained below. Otherwise the pixel is marked as "no-snow" (Tab. 2.6).

## 2.5 Snowline elevation

The algorithm works in two passes. The snow detection (Sect. 2.4.1) is performed a first time using parameters $n_1$ and $r_1$, which are set in such way as to minimize the number of false snow detections. This first pass enables to estimate the minimum elevation of the snow cover $z_s$, above which a second pass will be performed with less conservative parameters values $n_2$ and $r_2$. Based on regional analyses of the snowline elevation variability in mountain ranges, (e.g. Gascoin et al., 2015; Krajčí et al., 2016), we assume that large altitudinal variations of the snowline elevation are not likely at the scale of a tile of 110 km by 110 km. Therefore, the minimum snow elevation $z_s$ is considered uniform at the scale of the processed image.

Before proceeding to pass 2, the total snow fraction after pass 1 is computed. If this snow fraction is below $f_t$, then pass 2 is skipped as the sample of snow pixels is not considered as statistically significant to determine the snow line elevation. Otherwise, the pass 2 is activated. For that purpose, the DEM is used to segment the image in elevation bands with a fixed height of $d_z$. Then, the fraction of the cloud-free area that is covered by snow in each band (after pass 1) is computed. The algorithm finds the lowest elevation band $b$ at which the snow cover fraction is greater than a given fraction $f_s$. The value of $z_s$ is the lower edge of the elevation band that is two elevation bands below band $b$.

## 2.6 Cloud mask processing

The cloud mask in the input L2A product is conservative because (i) it is computed at a coarser resolution and (ii) it was developed to remove surface reflectance variations due to cloud contamination. However, the scattering of some thin clouds is low in the SWIR, green and red bands which are used for snow detection (Sect. 2.4.1). Hence, the human eye can "see" the





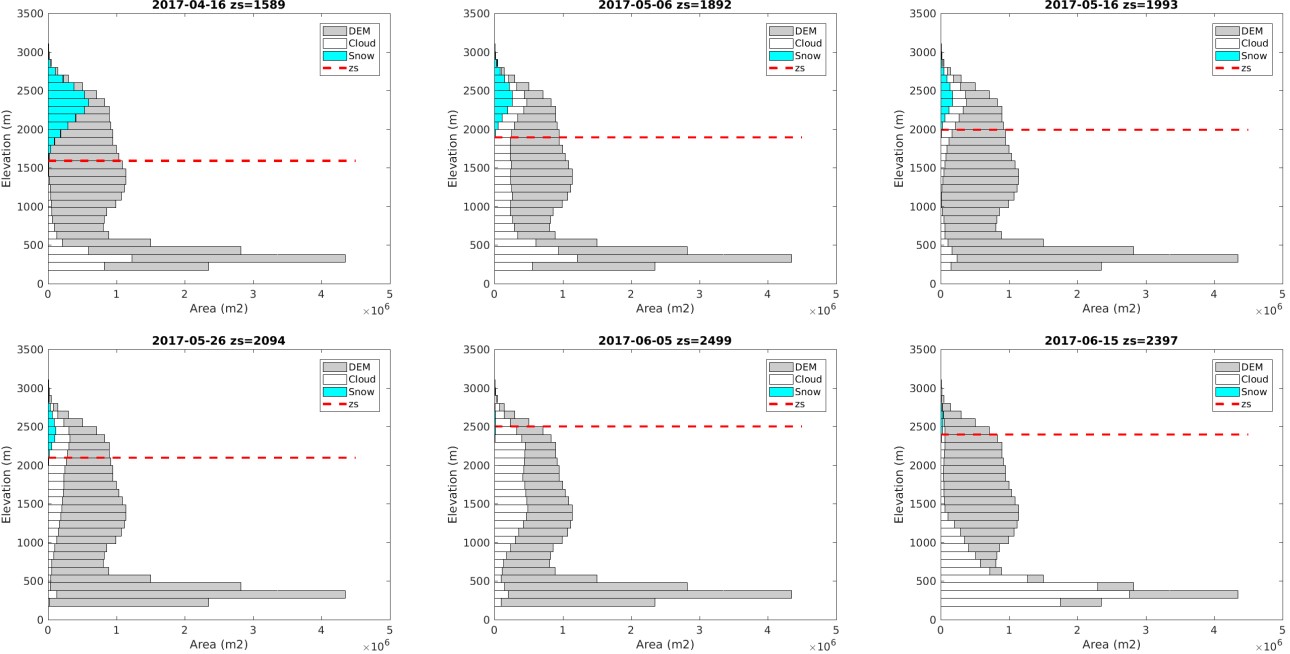

**Figure 1.** Time series of snow and cloud cover area distribution by elevation band over tile 31TCH (tile location in Fig. 5). The red dashed line indicates the position of $z_s$ as determined by the LIS algorithm. The corresponding snow maps are shown in Fig. 2.

snow (or the absence thereof) through these semi-transparent clouds in a color composite. In addition, the L2A cloud mask tends to falsely classify the edges of the snow cover as cloud. Therefore, the algorithm includes some additional steps to recover those pixels from the L2A cloud mask and reclassify them as snow or no-snow. This step is important because it substantially increases the number of observations as specified in Sect. 2.1.

A pixel from the L2A cloud mask cannot be reclassified as snow or no-snow if any of these conditions is satisfied:"

   – it is coded as "cloud shadow" in the L2A cloud mask;

   – it is coded as "high altitude cloud" (or "cirrus") in the L2A cloud mask;

   – it is not a "dark cloud" (see below).

The cloud shadows are excluded because the signal-to-noise ratio can be very low in these areas. The "high clouds" are
excluded because they can have a similar spectral signature as the snow cover, i.e. a high reflectance in the visible and a low reflectance in the SWIR. This type of cloud are detected in Sentinel-2 and Landsat-8 L2A products based on the spectral band centered on the 1.38 μm wavelength (Gao et al., 1993).

We select only the "dark clouds" as possible reclassification areas, because the NDSI test is robust to the snow/cloud confusion in this case. The "dark clouds" are defined using a threshold in the red band after down-sampling the red band by a factor
$r_f$ using the bilinear method. This resampling is applied to smooth locally anomalous pixels, following the MAJA algorithm,





**Figure 2.** Time series of snow maps over tile 31TCH (tile location in Fig. 5). The corresponding charts of snow and cloud cover area distribution by elevation band are shown in Fig. 1.

which performs the cloud detection at 240 m for L2A products (Hagolle et al., 2017). Therefore, if a (non-shadow, non-high-cloud) cloud pixel has a red reflectance at this coarser resolution that is lower than $r_D$, then it is temporarily removed from the cloud mask and proceeds to the pass 1 snow detection. The new cloud mask at this stage is the pass 1 cloud mask in Figure 3.

After passing the pass 1 and 2, some former cloud pixels (pixels that were originally marked as cloud in the L2A cloud
5    mask) will not be reclassified as snow because they did not fulfill the conditions of Equation 2.4.1. These pixels are flagged as cloud in the final snow product if they have a reflectance in the red that is greater than $r_B$. Otherwise they are classified as no-snow. Here the red band at 20 m resolution is used, to allow an accurate detection of the snow-free areas near the snow cover edges that the L2A product tend to falsely mark as cloud. The resulting cloud mask is the pass 2 cloud mask in Figure 3.





**Figure 3.** Flowchart of the snow detection algorithm. Table 2.6 gives the description and value of the algorithm parameters (written in red in this chart).



**Table 1.** LIS algorithm parameters description and default values for Theia's L2A Sentinel-2 (and Landsat-8) products.

| Parameter | Description | Value |
|---|---|---|
| $r_f$ | Resize factor to produce the down-sampled red band | 12 (8) |
| $r_D$ | Maximum value of the down-sampled red band reflectance to define a "dark cloud" pixel | 0.300 |
| $n_1$ | Minimum value of the NDSI for the pass 1 snow test | 0.400 |
| $n_2$ | Minimum value of the NDSI for the pass 2 snow test | 0.150 |
| $r_1$ | Minimum value of the red band reflectance the pass 1 snow test | 0.200 |
| $r_1$ | Minimum value of the red band reflectance the pass 2 snow test | 0.040 |
| $d_z$ | Size of elevation band in the DEM used to define $z_s$ | 0.100 |
| $f_s$ | Minimum snow fraction in an elevation band to define $z_s$ | 0.100 |
| $fc_t$ | Minimum clear pixels fraction (snow and no-snow) in an elevation band to define $z_s$ | 0.100 |
| $f_t$ | Minimum snow fraction in the image to activate the pass 2 snow test | 0.001 |
| $r_B$ | Minimum value of the red band reflectance to return a non-snow pixel to the cloud mask | 0.100 |

## 2.7 Implementation

The algorithm was implemented in an open source processor called let-it-snow (LIS). LIS is written in Python 2.7 and C++ and relies on the Orfeo Toolbox and GDAL libraries. GDAL is used for input/output operations, image resampling, and meta-data access (GDAL/OGR contributors, 2018). Orfeo Toolbox enables to make the computations with good performances under 5 memory constraints (i.e. the amount of available memory can be predefined), which is critical for operational production (Grizonnet et al., 2017). LIS takes as input a digital elevation model and a Theia L2A product from Sentinel-2 MSI, Landsat-8 OLI, SPOT-4 HRVIR, or SPOT-5 HRG (Gascoin et al., 2018). It is also compatible with Landsat-8 USGS level 2 products (U.S. Geological Survey Earth Resources Observation And Science Center, 2014) and Sentinel-2 ESA's sen2cor level 2A products (Louis et al., 2016).

## 10 3 Data description

The Snow Collection data can be freely accessed using a web browser by connecting to http://theia.cnes.fr or using this free command-line tool: https://github.com/olivierhagolle/theia_download. The user must first create an account in Theia to have the permission to download the data.

The Theia Snow collection is organized following the tiling system of Sentinel-2 (Fig.4). Each tile covers a square area 15 of 110 km by 110 km in the Universal Transverse Mercator (UTM) coordinate system. There are currently 127 tiles in the Snow collection, mostly over the mountain ranges of western continental Europe (France, Spain, Switzerland, Italy, western Austria). The snow products are also provided for the southern Quebec in Canada, the Issyk-Kul lake catchment in Kirghistan, the Kerguelen Islands. These extra-European sites were selected to support specific ongoing projects in relation with Theia.





**Figure 4.** Coverage of the Theia Snow collection tiles

An opportunistic tile is produced in Central Nevada, USA, because this tile is available as level-2A anyway for calibration purposes. The Theia Snow collection covers a number of mountain ranges with seasonal snow cover (Table 2).

The Theia Snow collection products are available for 127 tiles starting December 2017. At the time of writing there were over 17'000 available products. A set of 1300 products tagged as version 1.0 are available between November 2015 and June





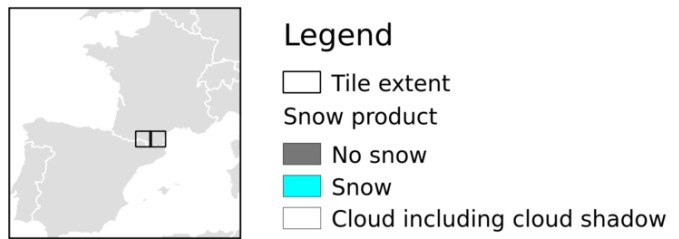

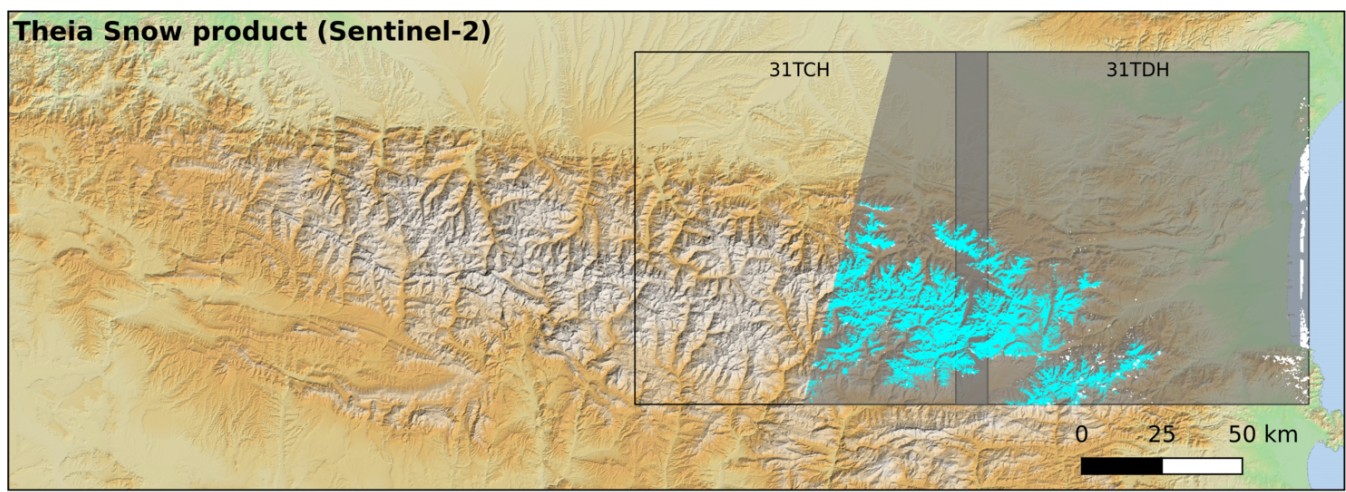

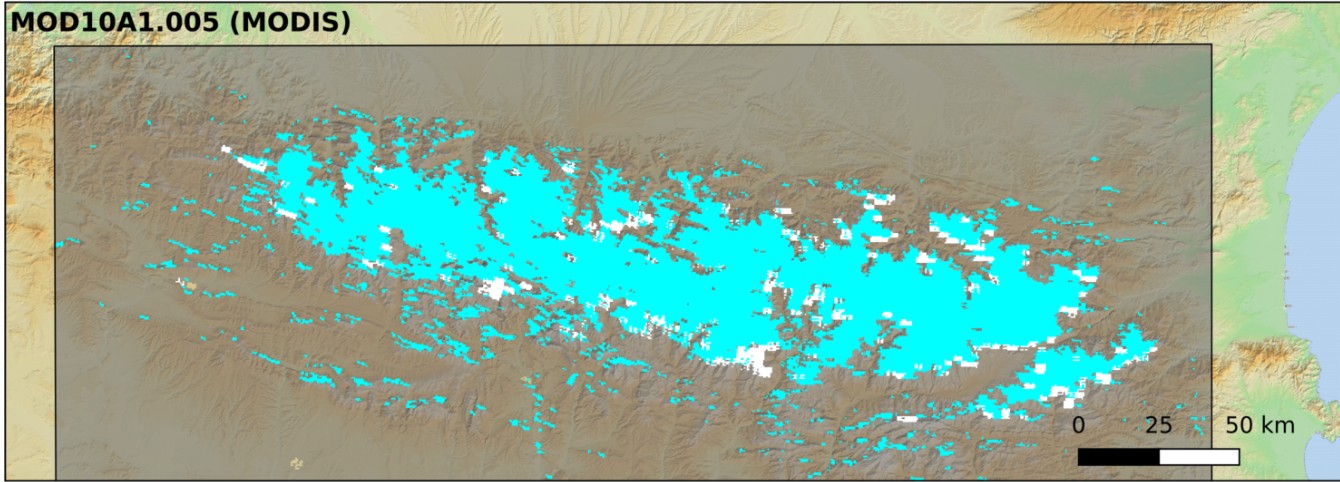

**Figure 5.** Map of the first two products in the Theia Snow collection and comparison with MOD10A1.005 snow product of the same day. The map background is the colorized version of the EU-DEM.





**Table 2.** Non exhaustive list of mountain ranges covered by the Theia Snow collection and corresponding tiles. The tiles marked with an asterisk were made available before the start of the operational production (see Sect. 3)

| Mountain range | Tiles |
| --- | --- |
| Alps | 32TMS,32TMR,32TMT,32TNS,32TNR,32TLQ*,32TLP,32TLS*,32TLR*,32TQT, 32TQS,33TUM,31TGK*,32TNT,31TGJ,32TPR,31TGL*,32TPT,32TPS |
| Appenines | 33TUH,32TNQ,33TUG,33TVG,33TWF,32TPP,33TWE,32TPQ,32TQP,32TQN |
| Cantabrian | 30TUN,29TPH,29TPG,29TQH |
| Corsica | 32TMM,32TMN,32TNM, 32TNM |
| Grant Range | 11SPC |
| High Atlas | 29RPQ*,29RNQ*,29SQR*,29SPR*,30STA* |
| Sistema Iberico | 30TVM,30TWK,30TWM,30TWL,30TXK,30TXL |
| Jura | 31TGM*,32TLT |
| Massif Central | 31TDL,31TDK,31TEK,31TEL |
| Pyrenees | 30TXN*,30TYM,30TYN*,31TCG,31TDG,31TCH*,31TDH* |
| Sierra de Cazorla | 30SWH |
| Sierra Nevada | 30SVG,30SWG |
| Sistema Central | 30TUL,30TUK,30TVL,30TTK |
| Vosges | 32TLT,32ULU |

These tiles can be viewed on an interactive map in this page https://theia.cnes.fr/atdistrib/rocket/#/sites

2017 for a subset of 15 tiles (Tab. 2). These products were produced in pre-operational using a different configuration of LIS, which underestimated the snow cover area in shaded slopes. All other products were generated with the same configuration as presented in this article. The products after December 2017 can still have a different version tag because of changing versions in the upstream L2A product. However, the different L2A versions have a low impact on the snow products. It

is expected that the cloud mask has improved gradually with time, due to algorithms enhancements in the successive L2A versions, and also because of the increasing revisit frequency of the Sentinel-2 mission (the full 5-day revisit became nominal above all land masses in March 2018). The different versions of the Snow Collection are listed and updated in this page: http://www.cesbio.ups-tlse.fr/multitemp/?page_id=13180.

The snow products are currently routinely generated at CNES using the MUSCATE scheduler, which also manages the L2A

10 production for Theia. MUSCATE performs a series of operations in high performance computing environment (i) download the level 1C product as soon it is available from the French mirror of the Sentinel products (PEPS, Plateforme d'Exploitation des Produits Sentinel), (ii) process and distribute the L2A product (iii) process and distribute the snow product. This workflow is optimized to reduce the lag time between the acquisition and the distribution of the products. As an example, in April 2018, nearly 80% of the snow products were made available online 5 days or less after the date of the Sentinel-2 acquisition (Fig. 6).

It is expected that this time lag should reach a median value of 2 days thanks to the increasing performance of MUSCATE.



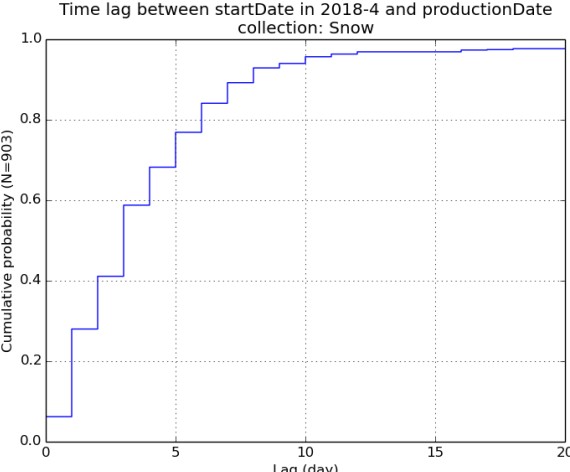

**Figure 6.** Cumulative probability of the time lags between the Sentinel-2 acquisition and the distribution date of the corresponding snow product, computed for all snow products with acquisition date in April 2018.

Each snow product is distributed as a zipped file which contains raster and vector files. The file base name is the product ID, i.e. `productId.zip`, where the `productId` is a character string which is constructed as follows:

`productId = Satellite_AcquisitionDate_L2B-SNOW_TileName_D_Version.`

For example, the eastern snow product in Fig. 5 was extracted from a file named:

`SENTINEL2A_20151130-105641-486_L2B-SNOW_T31TDH_D_V1-0.zip,`

which indicates that this product was generated from a Sentinel-2A image acquired on 30 November 2015 at 10:56:41.486 UTC and covers tile 31TDH. The version of the product is 1.0 (see above).

The size of each zipped product varies depending on the complexity of the snow and cloud mask, but generally ranges between 10 Mo and 100 Mo. After extracting the product the following files are created:

– `productID_SNW_R2.tif`: an 8-bit single band GeoTiff image which provides the following classification for each pixel:

–    0: no-snow,

–    100: snow,

–    205: cloud including cloud shadow,

–    254: no data.

–    `productID_SNW_R2.{shp,shx,dbf,prj}`: a vector file in the ESRI Shapefile format, which contains a vectorized version of the productID_SNW_R2.tif using the polygon geometry.





- `productID_CMP_R2.tif`: an 8-bit RGB GeoTiff image which shows the outlines of the snow mask (green lines) and cloud mask (magenta lines) on a false-color composite of the input L2A image (RGB image made with bands SWIR/Red/Green). This composition was chosen because RGB composites using the SWIR band image are useful to discriminate the snow cover from the snow-free areas and from the clouds (Vidot et al., 2017). Hence, this image mainly allows the user to visually check the consistency of the snow and cloud mask.

- `productID_EXS_R2.tif`: an 8-bit GeoTiff image for expert evaluation, where each bit indicates the value of an intermediate computation mask (Sect. 2):

    - bit 1: snow (pass 1),

    - bit 2: snow (pass 2),

    - bit 3: clouds (pass 1),

    - bit 4: clouds (pass 2),

    - bit 5: clouds (initial all cloud),

- `productID_MTD_ALL.xml`: a metadata file.

- `productID_QKL_ALL.jpg`: a quicklook picture of the `productID_CMP_R2.tif` image made using this colormap (8-bit RGB code in parentheses): snow in cyan (0,255,255), cloud in white (255,255,255), no-snow in grey (119,119,119) and no-data in black (0,0,0) (see an example in 7).

For most applications, only files with the `SNW` suffix should be useful.

The shapefile and raster images are referenced in WGS-84 UTM system with the zone number given by the first two digits of the tile name. All raster files have a 20 m resolution.

## 4 Evaluation

In this section we present an evaluation of the Snow collection. We first present the methods developed to do this evaluation and then the results.

### 4.1 Method

#### 4.1.1 Comparison with in situ snow depth measurements

We collected all available snow depth measurements within tiles 31TGM, 31TGL, 31TGK, 32TLS, 32TLR, 32TLQ, 31TDH, 31TCH, 30TYN and 30TNX from the Météo-France database between 01-Sep-2017 and 31-Aug-2018. These 10 tiles cover most of the french Alps and Pyrenees. We obtained 120 snow depth time series with at least one measurement. We gathered all available Sentinel-2 snow products for these tiles over the same period, i.e. a total of 1134 products. Then, we extracted

**Figure 7.** Example of a snow map (SNW file) and its corresponding color composite (CMP file). In the snow map the snow is represented in cyan, no-snow in grey and cloud in white. In the RGB composite, the snow cover areas are delineated in magenta while the clouds are delineated in green (Theia snow product of tile 32TNS on 27 May 2017).



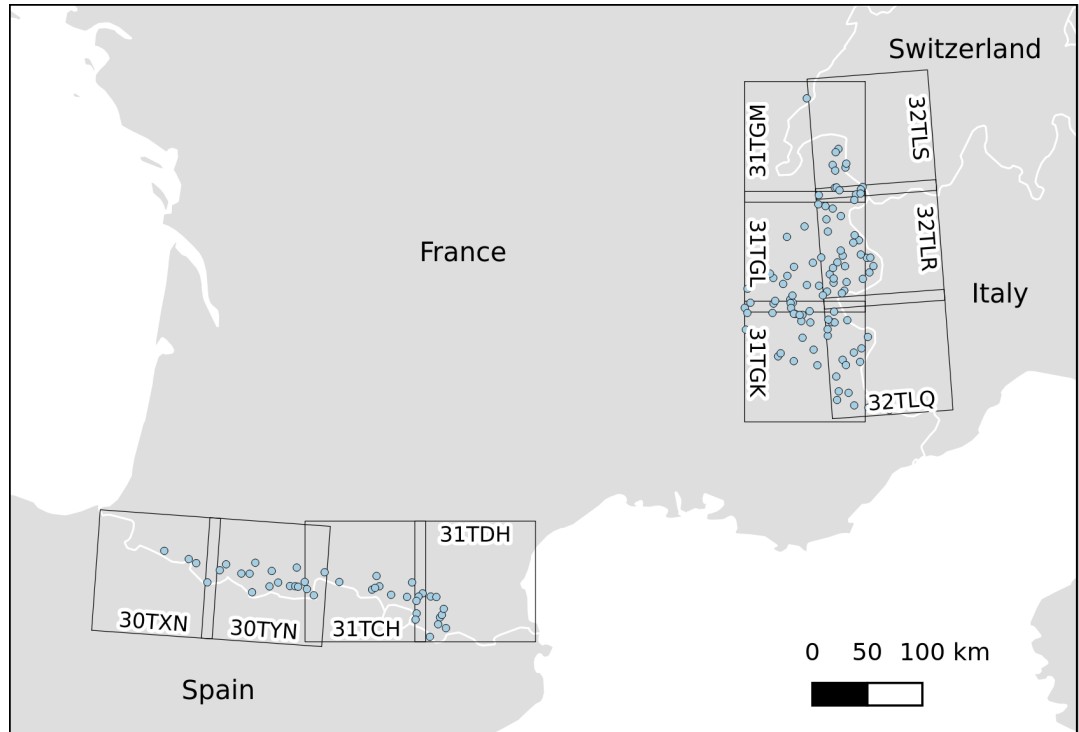

**Figure 8.** Map showing the location of the stations in the *réseau nivo-météorologique* used for this study (source: Météo-France) and the corresponding Sentinel-2 tiles.

the pixel values at the location of each snow measurement stations for all dates. When a station is located in two tiles we only kept the data from the first tile. We selected the snow depth measurements which were collected on the same day of a Theia snow product and discarded the measurements corresponding to a cloud detection in the Theia snow product. The snow depth measurements were converted to snow presence and absence using a threshold of $SD_0=0$ m. This eventually allowed us to compute a confusion matrix between a set of snow presence and absence data from in situ measurement and a set of simultaneous snow presence and absence data from the Theia Snow collection across the French Alps and Pyrenees. However, previous studies comparing MODIS binary snow products with ground measurements showed that a value of 0 m may not be optimal (Klein and Barnett, 2003; Gascoin et al., 2015), therefore the sensitivity of the results to $SD_0$ was tested by recomputing the confusion matrix for 1 cm increments of $SD_0$ from 0 to 1 m.

### 4.1.2 Comparison with snow maps of higher spatial resolution

We used SPOT-6 and SPOT-7 images with a resolution of 1.5 m in panchromatic and 6 m in multispectral (blue, green, red, near-infrared) to evaluate the accuracy of the snow detection in Theia snow products. For this comparison, the LIS processor was run with its current operational configuration (i.e. the configuration used to routinely produce the Theia snow products



**Table 3.** Pairs of Sentinel-2 and SPOT-6/7 products used for the evaluation of the snow detection accuracy (see also Fig. 3).

| Sentinel-2 product | | Reference product | |
| Tile | Date | Sensor | Date |
| --- | --- | --- | --- |
| 31TGK | 2016-12-11 | SPOT-7 | 2016-12-17 |
| 31TGL | 2017-03-11 | SPOT-6 | 2017-03-11 |
| 31TGL | 2016-12-01 | SPOT-7 | 2016-12-03 |
| 31TGK | 2017-03-11 | SPOT-6 | 2017-03-11 |
| 31TGK | 2016-08-13 | SPOT-6 | 2016-08-08 |
| 32TLS | 2016-10-12 | SPOT-7 | 2016-10-12 |

since December 2017 Tab. 2.6). SPOT-6 and SPOT-7 sensors acquire images with a large radiometric depth coded in 16 bits, which enables to identify surfaces features in alpine regions with dark shaded slopes and bright snow surfaces. We searched in the catalogue of the Kalideos database for available SPOT images that could match a cloud-free (or nearly cloud-free) Theia snow map over the French Alps region (https://alpes.kalideos.fr). We identified six pairs of images in 2016 and 2017 with a

maximum time lag of 6 days (Tab. 3, Fig. 9). The SPOT images were obtained as orthorectified products from the Kalideos database. The SPOT images were used to generate reference snow maps by a pixel-based supervised classification. For each SPOT image, we manually drew about fifteen polygons of homogeneous of snow and no-snow surfaces using the panchromatic image. The cloud cover in the SPOT images was also manually delineated with a large buffer to restrict the classification to strictly cloud-free areas. Color composites made from the multispectral images were also used as a visual support to help the

snow and cloud classification. These polygons were then used to extract training samples from the SPOT products. The samples were taken from both the panchromatic and the multispectral image. the We tested a number of classification algorithms by splitting the polygons in calibration and validation datasets. The detail of this analysis is not shown here, but it allowed us to conclude that the random forest classifier was the best choice for our purpose, given its good accuracy and its numerical efficiency (Bouchet, 2018).

In addition, we generated the snow maps from the same Sentinel-2 level 1 products using ESA's sen2cor version 2.5 (Louis et al., 2016). The sen2cor output snow masks are expected to be nearly identical to the ones included in the distributed L2A products by ESA since the same processor is used. We have generated ourselves the L2A products with sen2cor for this study, because the L2A products were not yet available when me made this analysis (ESA started operational production of Sentinel-2 data at level 2A in 2017). The Theia and sen2cor snow maps were compared to the reference snow maps from SPOT images

using standard metrics derived from the confusion matrix (accuracy, f1-score, kappa, false detection rate and false negative rate).



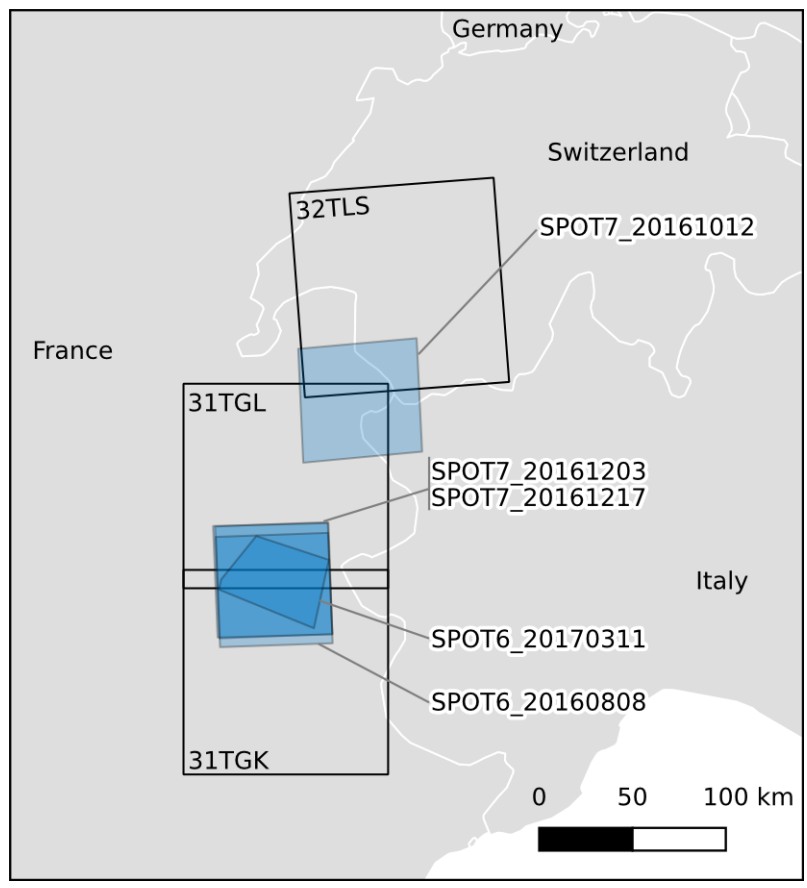

**Figure 9.** Location of the five SPOT-6/7 products used for the evaluation of the snow detection accuracy and the corresponding Sentinel-2 tiles (see also Tab. 3).

### 4.1.3 Visual verification

The evaluation methods presented above under-sample the actual resolution of the Theia snow collection both in space (only 120 points of comparisons in Sect. 4.1.1) and time (only 6 dates of comparisons in Sect. 4.1.2). Given that we do not have a more extensive validation dataset, we used a time series of 64 Theia snow products from 06 July 2015 to 02 July 2017 over tile

5  31TCH to control the consistency of the snow and cloud masks based on the visual inspection of the false color composites. This approach is efficient to evaluate the frequency of gross errors at the tile scale, i.e. large patches of false snow or false no-snow detection (Vidot et al., 2017).



**Table 4.** Confusion matrix between the Theia snow products and in situ snow depth data (SD).

|  | Theia snow products | |
| --- | --- | --- |
| In situ snow depth | no snow | snow |
| SD=0 | 276 | 8 |
| SD>0 | 76 | 1054 |

## 4.2 Results

### 4.2.1 Comparison with in situ snow depth measurements

The confusion matrix between the Theia snow products and in situ snow depth measurements is given in Table 4. We find 1414 pairs of data for the study period (i.e. there are 1414 individual snow depth measurements for which a cloud-free retrieval

can be found in the Theia Snow collection on the same day). From this confusion matrix, the accuracy (proportion of correct classifications) is 94% and the kappa coefficient (Cohen, 1960) is 0.83, which indicates a excellent agreement according to Fleiss et al. (2013). The false positive rate (2.8%) is lower than the false negative rate (6.7%), which means that the Theia snow product is more likely to underestimate than to overestimate the snow detection at the station locations. The false negative rate decreases if $SD_0$ is set to higher value, however the false positive rate also increases in such a way that an optimum is reached

at $SD_0 = 2$ cm (Fig. 10).

The comparisons of each individual time series before matching the data by date is shown in Appendix B. These plots illustrate that the high revisit time of Sentinel-2 enables to well capture the seasonal cycle of the snow cover. Even intermediate melt out events at lower elevation stations can be identified over the course of the snow season.

### 4.2.2 Comparison with snow maps of higher spatial resolution

Table 5 shows that the MAJA-LIS workflow provides a better detection of the snow cover than the sen2cor processor in all the studied cases. Although the improvement in the accuracy coefficient is low in some cases (31TGL/2017-03-11, 31TGK/2016-08-13) or even slightly negative in one case (32TLS/2016-10-12), the F1 score and kappa coefficient of the Theia snow products are always greater than or equal to those of sen2cor. The differences are significant for four products among the six tested. The lower accuracy in the case of the the 32TLS/2016-10-12 product is due to a higher false negative rate. The false positive rates

in the Theia snow products are generally lower than the false negative rates, which is consistent with the previous evaluation using in situ data. The spatial analysis of the errors indicates that the false positives (i.e. over-detection of snow pixels) in the Theia products are mostly located near the snow cover edges (Fig. B4). This suggests that these errors might not be due to a false detection of snow (e.g. confusion with a lake or a cloud surface) but are rather due to the resolution discrepancy with the reference data. By contrast, the spatial comparison of the sen2cor products with the reference data shows large patches of false



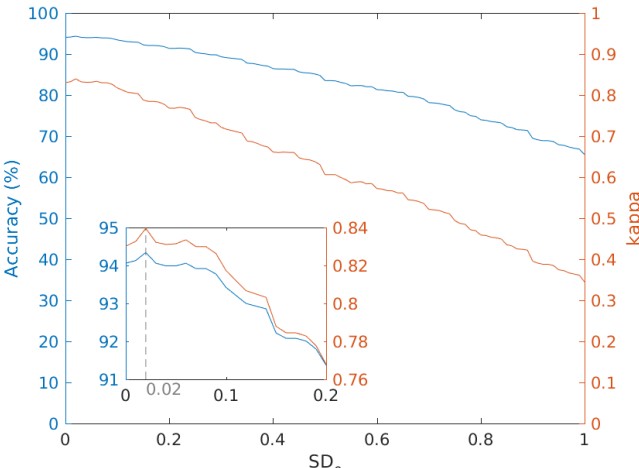

**Figure 10.** Sensitivity of the agreement between in situ and Theia snow product to $SD_0$ in m (threshold to convert the measured snow depth to snow presence or absence). The inset shows a zoom in the region 0-0.2 m.

**Table 5.** Results of the validation of Theia (LIS 1.2.1) and ESA (sen2cor 2.5) snow products using SPOT-6/7 reference snow maps.

| Sentinel-2 product | | Accuracy | | F1 | | Kappa | | FPR | | FNR | |
|---|---|---|---|---|---|---|---|---|---|---|---|
| Tile | Date | sen2cor | LIS | sen2cor | LIS | sen2cor | LIS | sen2cor | LIS | sen2cor | LIS |
| 31TGK | 2016-12-11 | 0.83 | 0.92 | 0.73 | 0.91 | 0.61 | 0.84 | 0.41 | 0.04 | 0.05 | 0.14 |
| 31TGL | 2017-03-11 | 0.89 | 0.90 | 0.92 | 0.93 | 0.74 | 0.76 | 0.09 | 0.02 | 0.06 | 0.11 |
| 31TGL | 2016-12-01 | 0.84 | 0.95 | 0.78 | 0.95 | 0.66 | 0.90 | 0.35 | 0.05 | 0.01 | 0.06 |
| 31TGK | 2017-03-11 | 0.85 | 0.89 | 0.89 | 0.92 | 0.65 | 0.72 | 0.15 | 0.01 | 0.07 | 0.13 |
| 31TGK | 2016-08-13 | 0.98 | 0.99 | 0.35 | 0.77 | 0.34 | 0.77 | 0.78 | 0.24 | 0.22 | 0.22 |
| 32TLS | 2016-10-12 | 0.98 | 0.97 | 0.63 | 0.64 | 0.62 | 0.62 | 0.49 | 0.08 | 0.17 | 0.51 |

negative (omission of snow pixels) in the snow cover area, which are typically due the omission of snow pixels in shaded areas (Fig. A2).

### 4.2.3 Visual verification

We found 4 products among 64 with significant patches of false positive pixels (i.e. pixels falsely classified as snow). These
5 false snow areas are exclusively located in cloud areas (Fig. 11). False negative pixels (i.e. pixels falsely classified as snow) can be found by looking at higher resolution along the snow cover edges, but we did not find large areas of false negative pixels.



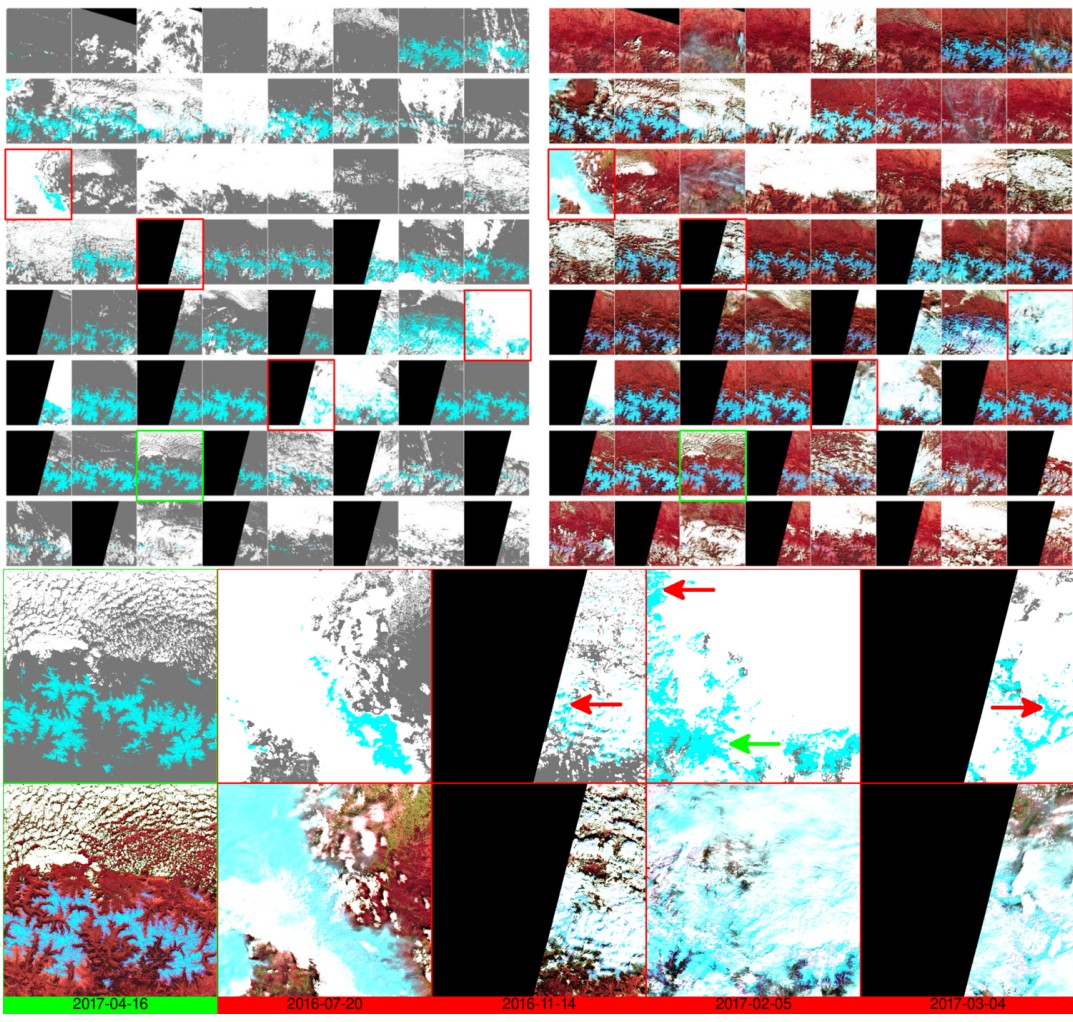

**Figure 11.** Full time series of Theia snow products from 06 July 2015 to 02 July 2017 over tile 31TCH. Four products with visually-evident false snow detections are framed in red and shown at higher resolution. Red arrows indicate the location of the false snow patches, while the green arrow indicates a true snow patch. The product of 16 April 2017 is shown as a reference to help the visualization of the errors.





## 5 Discussion

The evaluation of the Snow collection using both in situ and remote sensing reference datasets indicates a excellent accuracy of the snow detection, albeit with a tendency to snow under-detection. The omission of snow pixels can be due to the presence of trees or shadows. In the case of the in situ comparison, part of the errors may be explained by the uncertainty in the geo-localization of the Sentinel-2 data, which is close to 11 m at 95% for both satellites according to the latest Sentinel-2 L1C data quality report (MPC Team, 2018). In addition, in situ measurements may not representative of the snow conditions within the Sentinel-2 pixel. However, the sensitivity analysis of the accuracy and kappa coefficient to the $SD_0$ value suggests that the discrepancy between the scale of the in situ observations and the scale of the pixel observation is not significant since the optimal $SD_0$ is close to 0 m. In contrast, a similar analysis with lower resolution snow products from MODIS found an optimal $SD_0$ of 0.15 m. This illustrates that high resolution snow products from Sentinel-2 enable to partly resolve the typical scale issue between in situ and remote sensing products Blöschl (1999).

The visual inspection of the products reveals that the main issue in some Theia snow products is rather the occurrence of false positive due to the confusion with cloud surfaces. This problem can be due to two main reasons:

– Cold clouds: cold clouds have a spectral signature that is close to the snow cover since they contain ice crystals (high NDSI). If these clouds are not accurately detected by the level 2A processor then the LIS algorithm will also classify them as snow after pass 1 (Sect 2). As a result the snowline elevation $z_s$ is not well estimated, which can generate erroneous snow patches within these clouds even at low elevation (e.g. product of 20 July 2017 in Fig. 11).

– Shaded clouds: the three-dimensional structure of the clouds can form shadows within the cloud cover area (e.g. product of 04 March 2017 in Fig. 11). These shaded cloud areas have a lower reflectance in the visible wavelengths, and therefore can be considered as "dark clouds" in the LIS algorithm, provided that the shaded cloud cover area is large enough to significantly reduce the reflectance in the red band even after the down-sampling step (Sect 2).

Among 64 products, we find 4 cases with significant false snow cover detection in this tile (Fig 11). Although it is not shown here, we have made this exercise over other tiles in the Alps and Atlas mountains and we estimate that this proportion is similar in these areas. Regarding this issue we mainly rely on our visual judgment using color composite to assess the ability of the algorithm to discriminate the cloud and of the snow cover. However, to further quantify the algorithm performance and eventually optimize its parameters, we would need a database of classified imagery with labelled regions of cloud and snow cover in many different cases.

## 6 Conclusions

The Theia Snow collection is a free collection of snow products which indicate the presence or absence of snow on the land based on Sentinel-2 (20 m) and Landsat-8 (30 m) data. Most of the snow products are derived from Sentinel-2 because the integration of Landsat-8 data started during the writing of this manuscript (October 2018). However, the processing is done





with the same algorithm, which is based on the calculation of the NDSI from slope-corrected surface reflectance (level 2A data). The cloud mask is also taken from the Theia level 2A data, although the algorithm allows marginal corrections of the cloud mask to reduce the number of cloud pixels.

The evaluation presented in this article indicate that the Theia Snow product allows an accurate detection of the snow
cover (and of the absence of snow). However, it remains a preliminary assessment that should be extended in the future. For example, the evaluation was based only on Sentinel-2 products, because the production for Landsat-8 had just started at the time of writing and thus the Landsat-8 products represented a very minor fraction of the Theia Snow collection. In addition, the evaluation was limited to the French Alps and French Pyrenees, while the collection covers other mountain regions. Last, the spatial evaluation using higher resolution remote sensing data was focused on the accuracy of the snow detection using
very high resolution clear-sky images, while, as discussed above, the main difficulty is not to detect the snow when the sky is cloud-free, but rather to avoid false snow detection within clouds. The reduction of the false positive rate is the main challenge for the future developments in the LIS algorithm. In the meantime, we also welcome feedback from other users.

The Theia Snow collection is based on optical observations therefore it is not adapted to the detection of the snow cover in dense forest areas where the ground is obstructed by the canopy (Xin et al., 2012). This may typically occur in evergreen
conifer forests of alpine regions (e.g. Alps, Pyrenees). We have noted that the snow detection can be successful even in alpine forests, but we lack quantitative data to quantify its accuracy. Therefore it is recommended to use a land cover map to exclude these forest areas from the analysis. For the tiles in France, this can be done using the Theia land cover map (Inglada et al., 2017), which is distributed at the same spatial resolution. In the Pyrenees, the mean altitude of zero-degree isotherm in winter is close to the treeline elevation at 1600 m, therefore the impact of the forest cover is limited (Gascoin et al., 2015). The LIS
algorithm may also fail to detect the snow cover in steep shaded slopes if the solar elevation is very low (typically below 20°). This can occur in mid-latitude areas in winter. In this case the slope correction in the L2A product is generally not applied. This is indicated in the L2A mask but not in the snow product. A future release of the Snow collection should include this information. However, the visual inspection of many images in regions of complex topography gives us the impression that the snow detection algorithm performs already well in shaded slopes. We are planning to further investigate this issue using
alternative approaches (Sirguey et al., 2009).

In spite of all these limitations, the Theia snow products have already been successfully used for the evaluation of the MODIS snow products (Masson et al., 2018) and the assimilation of snow cover area data into a snowpack model in the High Atlas (Baba et al., 2018). Given that the snow cover is a key driver of many natural processes in mountains regions, we envision various potential applications of the Theia snow products, including the modelling of the distribution of the permafrost in mountain
regions, the validation and calibration of hydrological models in snow-dominated catchments, and the spatial modelling of biodiversity and productivity of ecosystems in mountain regions.



## 7 Code and data availability

The Theia snow collection can be accessed and cited using this Digital Object Identifier: DOI:10.24400/329360/F7Q52MNK. The let-it-snow source code is available under the GNU Affero General Public License v3.0 in this repository: http://tully. ups-tlse.fr/grizonnet/let-it-snow.



**Appendix A:  Spatial comparison between Theia and sen2cor snow products with SPOT-6/7 reference data**

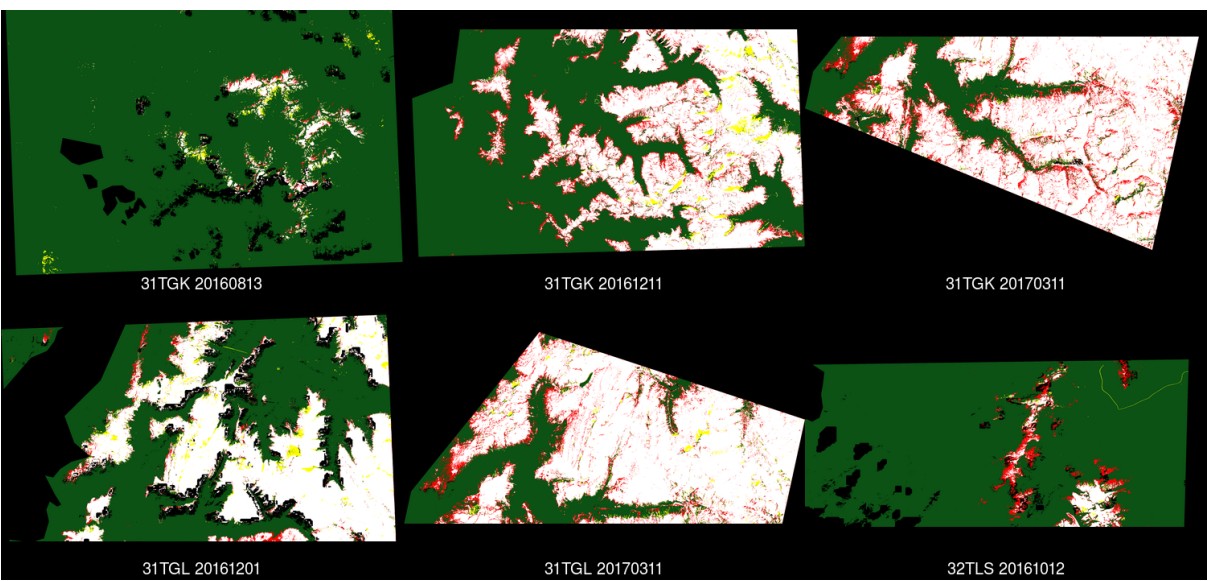

**Figure A1.** Theia snow products vs. SPOT-6/7 snow maps (green: true negative, white: true positive, red: false positive, yellow: false negative).

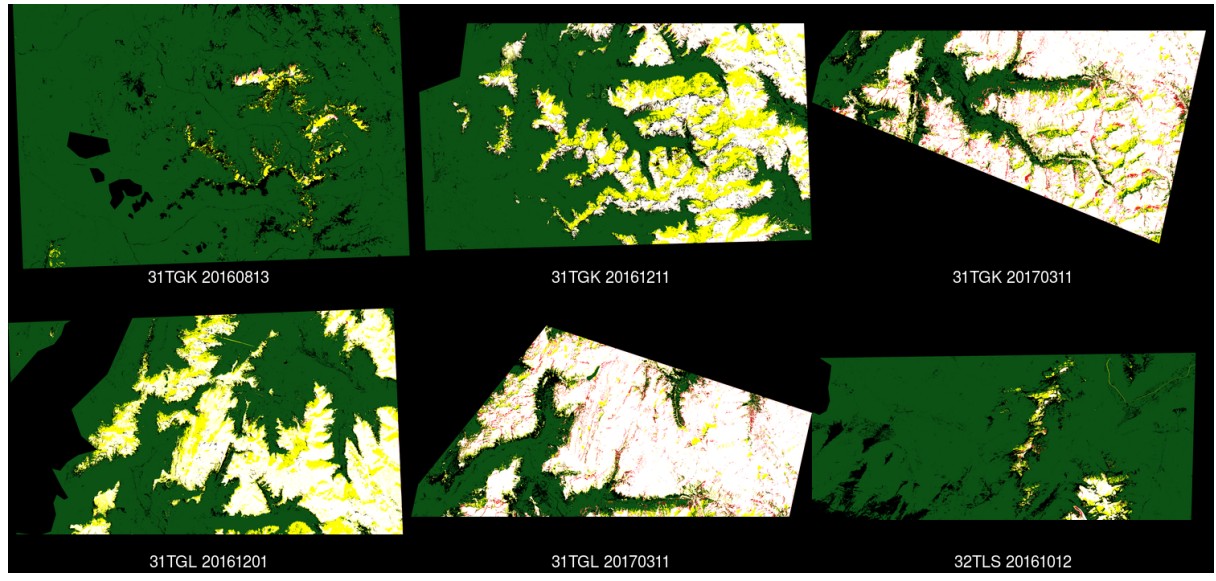

**Figure A2.** Sen2cor snow product vs. SPOT-6/7 snow maps (green: true negative, white: true positive, red: false positive, yellow: false negative).




**Appendix B:  Time series of Theia snow products and snow depth records at 120 snow observation stations from 01-Sep-2017 to 31-Aug-2018.**

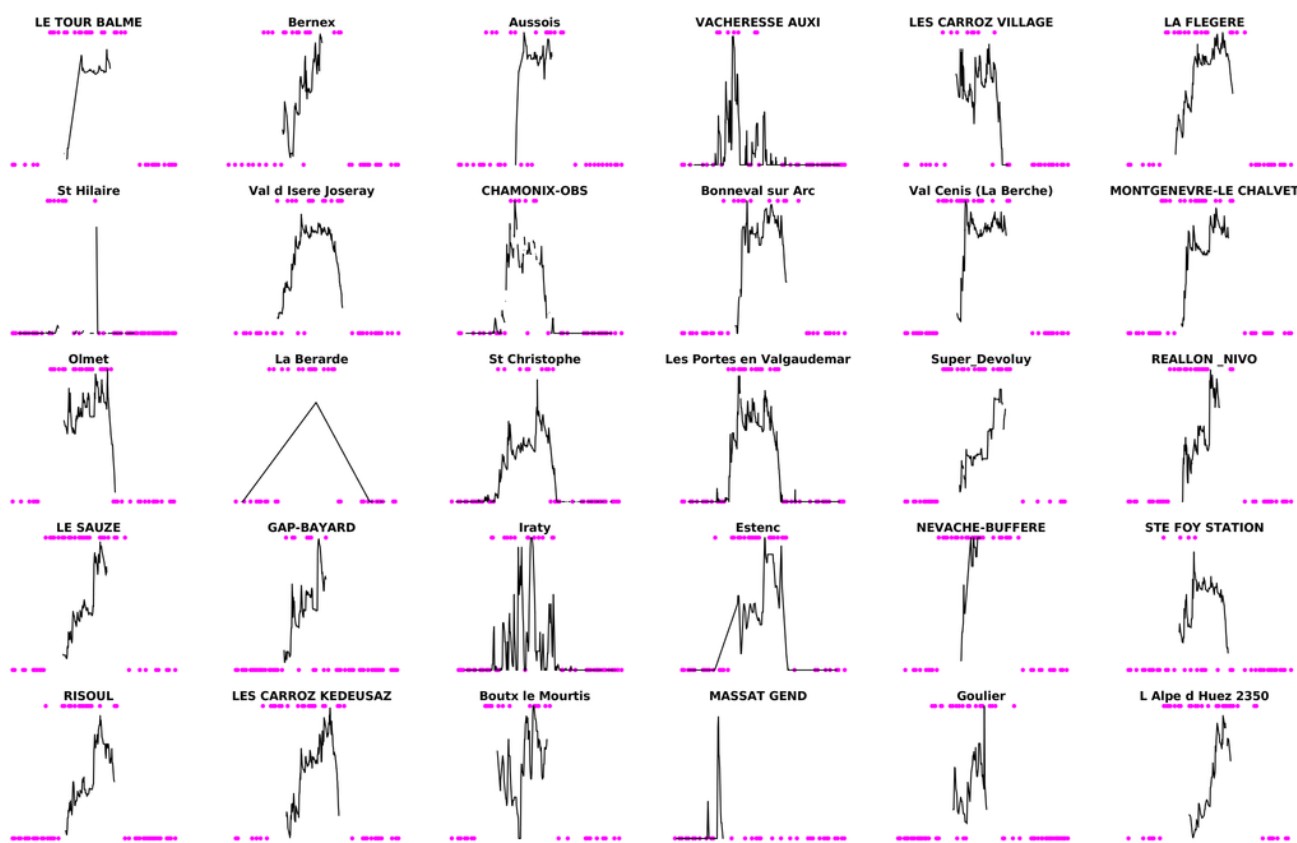

**Figure B1.** Magenta dots: snow presence (top) or absence (bottom) in the Theia snow collection. Black line: snow depth time series.





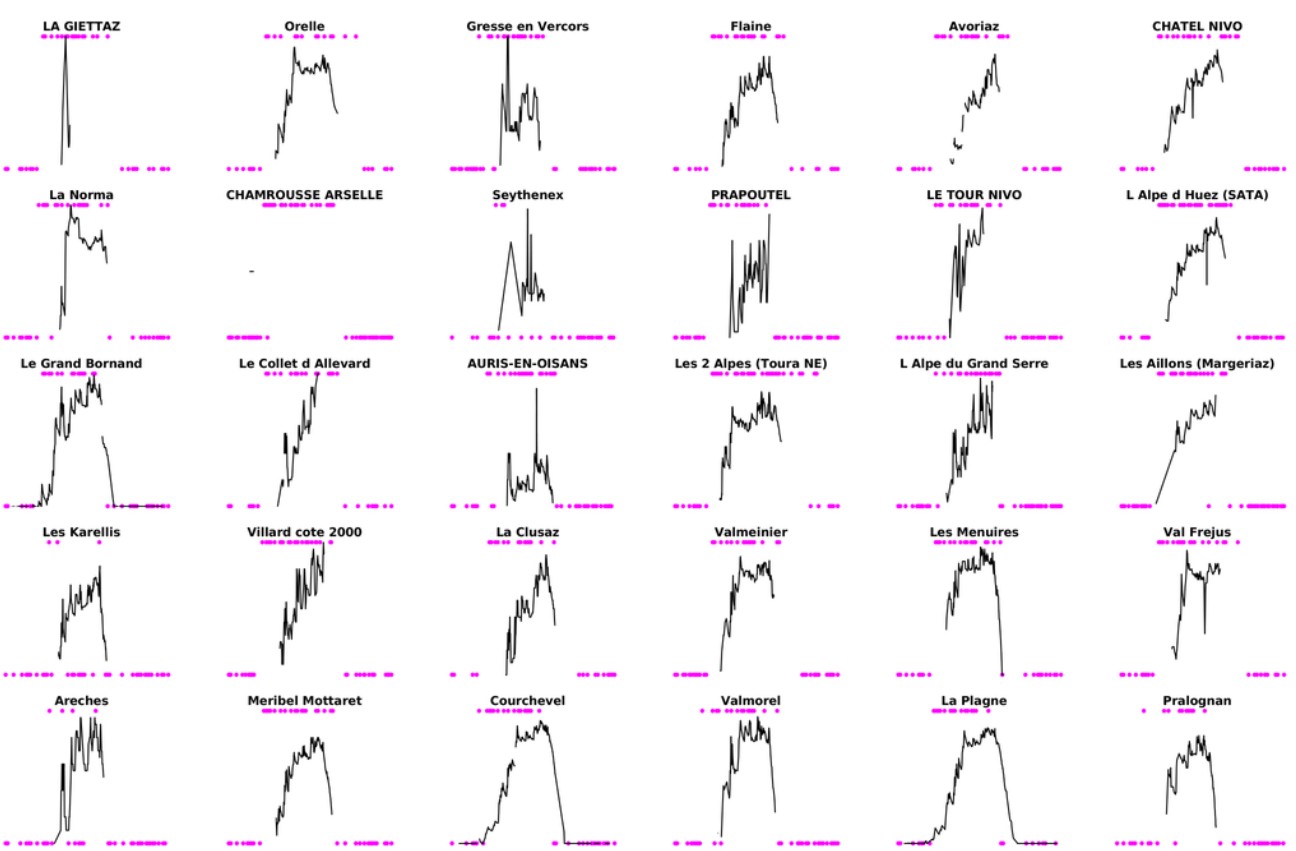

**Figure B2.** Magenta dots: snow presence (top) or absence (bottom) in the Theia snow collection. Black line: snow depth time series.





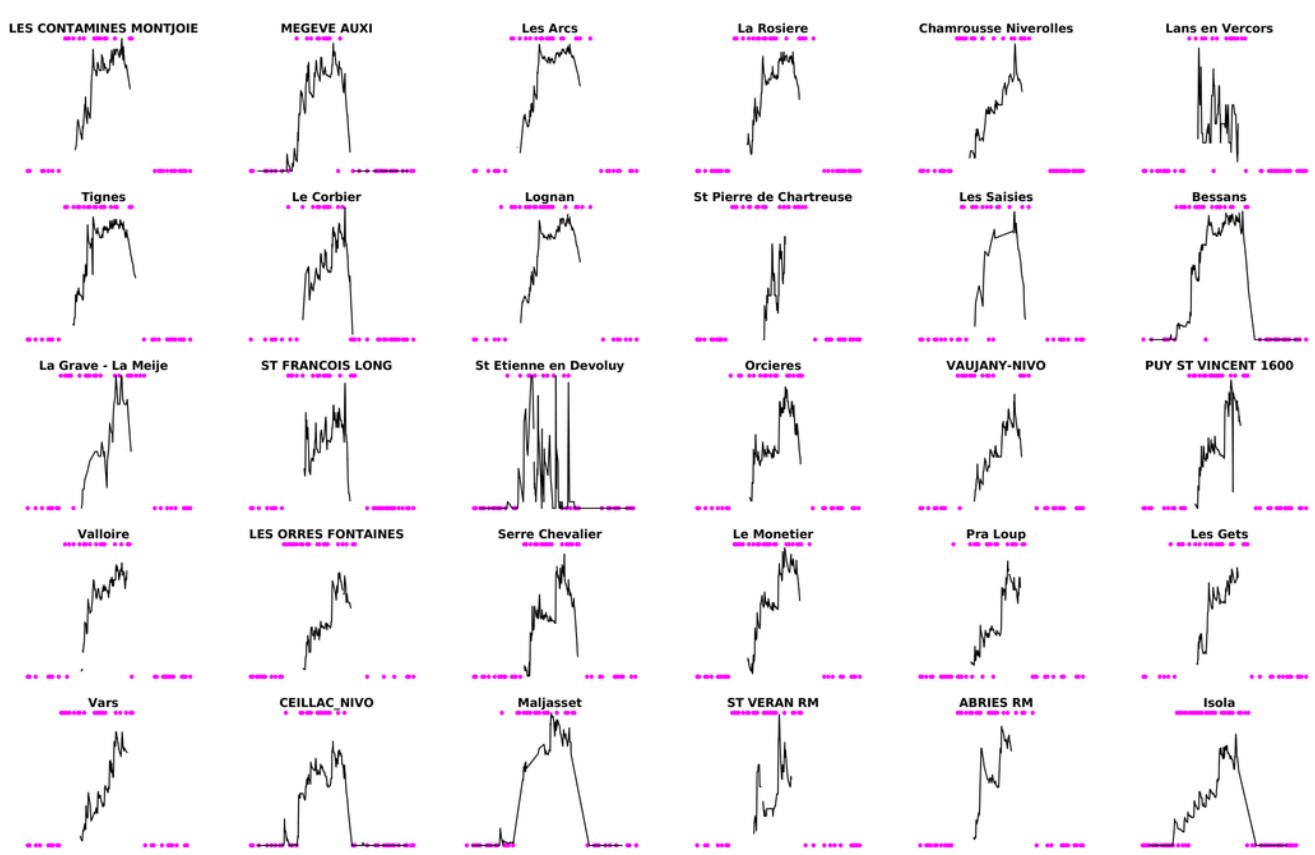

**Figure B3.** Magenta dots: snow presence (top) or absence (bottom) in the Theia snow collection. Black line: snow depth time series.



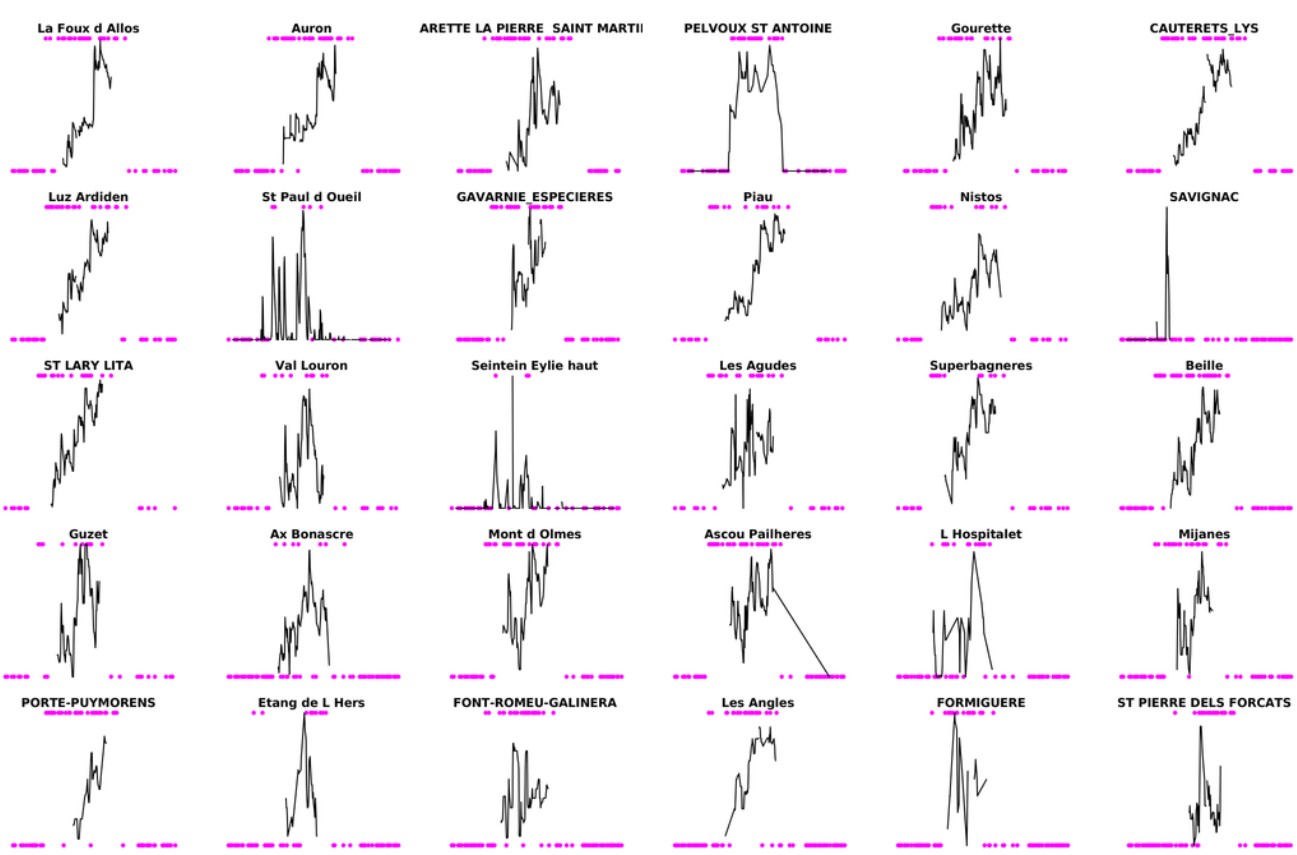

**Figure B4.** Magenta dots: snow presence (top) or absence (bottom) in the Theia snow collection. Black line: snow depth time series.




*Author contributions.*  SG and OH defined the LIS algorithm. MG and GS worked on the implementation and developments of the LIS processor. MG, MB and SG worked on the validation of the snow products. SG wrote the manuscript with inputs from all the co-authors.

*Competing interests.*  The authors declare that no competing interests are present.

*Acknowledgements.*  The development and production of the Theia Snow collection is supported by the CNES. We thank Tristan Klempka
5   for his contribution to earlier developments of the LIS processor. We thank Marc Leroy, Arnaud Sellé and Philippe Pacholczyk for the coordination of the Theia project, Joelle Donadieu and Céline l'Helguen for the transfer to the production at MUSCATE.



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
