# Peer review of "Theia Snow collection: high resolution operational snow cover maps from Sentinel-2 and Landsat-8 data"

_Earth System Science Data, 2018_

## Referee Comment (RC1) · Piazzi (Referee) · 30 Dec 2018

**[essd-2018-144] - Theia Snow collection: high resolution operational snow cover maps from Sentinel-2 and Landsat-8 data**

Please note that text citations are reported in italics.

**GENERAL COMMENTS**

The authors introduce a freely-available collection of snow products derived from Sentinel-2 and Landsat-8 data through a robust retrieval algorithm, which allows overcoming well-known limitations affecting the snow detection. The newly-developed algorithm is described in detail and its potential is tested and discussed through several analyses. The satellite-based dataset presented in this study is of critical importance and great usefulness not only for the scientific research community, but also for users interested in operational applications.

I have identified some issues which deserve only minor revisions of the manuscript before its publication.

**SPECIFIC COMMENTS**

Here are listed my main comments:

1. The manuscript contains several self-explanatory figures, which make it easier to read and understand the text. However, further references to figures need to be introduced, since for some of them the reference is even missing.
2. The manuscript comprehensively describes the newly-developed algorithm and the accuracy analysis of the resulting satellite products. However, a description of the satellite datasets used in this study (i.e., Sentinel-2 and Landsat-8) is missing. It would be of key interest to introduce the spectral bands of these satellite imagery to support the description of LIS algorithm.
3. A further section introducing the evaluation metrics should be added to better enable the assessment of the results.
4. Although the title mentions both Sentinel-2 and Landsat-8 data, references to Landsat-8 dataset are generally missing in the manuscript. Indeed, both methodology and results are actually referred to Sentinel-2 imagery.
5. The final section 6 includes information of critical importance, which should be reported and discussed in the text. Please consider my comments on Section 6.

Please consider replacing *sen2cor* with *Sen2Cor* throughout the whole text.

**Section 1 (Introduction)**

Pag. 1, l. 20: Please report the reference within brackets.

Pag. 1, l. 22: Please replace *12h* with 12 hours.

Pag. 2, l. 5: Please consider reporting the full name of the plan.

Pag. 2, l. 10: Please report the reference within brackets.

Pag. 2, l. 14: Please consider replacing *5 day* with 5-days.

Pag. 2, l. 19, 21: Please explicitly define and explain what level-1C and level-2A products are (i.e., TOA, BOA).

Pag. 2, l. 21: Please specify the starting sensing date of the processed L2A product provided by ESA.

Pag. 2, l. 24: What are the main limitations of ESA's L2A algorithm for snow detection?

Pag. 2, l. 28: Please consider explicitly specifying here that Theia collection relies on a combination of these two satellite datasets. Currently this information is more clearly reported only in the Conclusions (see main remark n°5 in General comments).

Pag. 2, l. 28: Please consider rephrasing the sentence as "… at 20-m and 30-m resolution, respectively…".

Pag. 2, l. 30: Please report here the reference for the Normalized Difference Snow Index.

Pag. 2, l. 31: Please close the bracket.

**Section 2 (Algorithm)**

Please add in the text references to Figures 1 and 2, which need to be commented (see main remark n°1 in General comments).

Please refer to Table 1 whenever an algorithm parameter is introduced.

Generally, in Section 2 the authors do not explain how the parameters values have been defined and optimized. Please add this missing information in the algorithm description.

**Section 2.1 (Scope)**

Currently some limitations of the ESA's processor are discussed within the description of LIS algorithm, for instance the critical issues affecting the cloud mask (i.e., thin clouds, edges of snow cover). In order to improve the readability of the manuscript and to properly highlight the potential of LIS algorithm, the authors could clearly identify and explain in this section what the main limitations of the ESA's processor are, before describing their algorithm.

**Section 2.2 (Input)**

In this section the input data are listed, but the data processing to derive input *L2A product* is not discussed. This section should also briefly explain how the correction of atmospheric and terrain slope effects is performed.

Pag. 3, l. 13: Please consider replacing *remote sensing images* with *remotely-sensed images*.

**Section 2.3 (Pre-processing)**

Pag. 3, l. 22: Please specify the original spatial resolution of the red and green bands. It would be useful to introduce a table reporting the spectral bands and their central wavelengths of the original satellite data (see main remark n°2 in General comments).

Pag. 3, l. 23: Please spell out *SRTM*.

**Section 2.4.1 (Snow detection)**

Are the cloud-contaminated pixels preliminary flagged and neglected in this first procedure? If so, please specify this information.

**Section 2.5 (Snowline elevation)**

Please consider renumbering this Section as 2.4.2.

Pag. 4, l. 15: How are the false snow detections defined? What is the reference dataset to identify false snow detections?

Pag. 4, l. 18: Sentinel-2 tile is 100 $km^2$.

Pag. 4, l. 20: Please specify how snow fraction is evaluated (fraction evaluated over the total number of snow and no-snow pixels?). Furthermore, how is defined the threshold of snow fraction $f_t$?

Pag. 4, l. 23-24: How are defined $d_z$ and $f_s$?

**Section 2.6 (Cloud mask processing)**

Please consider renumbering this Section as 2.4.3.

Pag. 5, l. 15: Please either specify how factor $r_f$ is defined or neglect this information by just reporting the corresponding reference.

Pag. 6, l.  2, 6: Please specify how $r_D$ and $r_B$ are defined.

Figure 3: Please introduce MUSCATE in text before this figure.

**Section 2.7 (Implementation)**

Please consider renumbering this Section as 2.4.4.

**Section 3 (Data description)**

Pag.  9, l. 3: The number of selected tiles is already reported in pag. 8, l. 15.

Pag. 12, l. 12: Please specify how the pixels classified as no-snow are defined. Are all snow- and cloud-free pixels classified as no-snow ones? Does the no-snow class include water pixels? Here it should be also mentioned the uncertainty in classifying vegetated pixels as snow or no-snow ones.

Pag. 12, l. 13: Please specify what pixels are classified as no-data.

Please switch Figure 5 and 6 since they are mentioned reversely in the text.

**Section 4.1 (Method)**

Since in this section the authors refer only to Sentinel-2 (e.g., pag. 13, l. 28; Table 3), they should explicitly specify that the evaluation is based only on Sentinel-2 data, as clearly stated in Section 6 (see main remark n°5 in General comments).

As this manuscript is focused on the mapping of snow cover, it would be more proper to limit the analysis over the snow period, not up to the end of summer season. For instance, Table 3 reports a selected pair of snow products in August, which is not of high interest for the main purpose of this research. Furthermore, it is noteworthy that, when assessing the contingency matrix, the analysis of these snow-free scenes could affect the representativeness of the results, as shown by the significantly higher number of correct negatives in Table 4.

**Section 4.1.1 (Comparison with in situ snow depth measurements)**

Please add in the text references to Figures 7 and 8, which need to be commented (see main remark n°1 in General comments).

Pag. 15, l. 2: How is defined the *first tile*? Are the Sentinel-2 data of the second tile neglected even when sensed in a different day with respect to the selected first tile? If so, it would entail a loss of useful remotely-sensed snow data.

**Section 4.1.2 (Comparison with snow maps of higher spatial resolution)**

Pag. 16, l. 11: Please remove *the*.

Pag. 16, l. 11: Could the authors mention the classification algorithms that have been tested and provide their references?

**Section 4.2.1 (Comparison with in situ snow depth measurements)**

To improve the assessment of the results, please report how the evaluation metrics are defined (see main remark n°3 in General comments).

**Section 4.2.2 (Comparison with snow maps of higher spatial resolution)**

While in Section 4.1.2 the authors explain that "*The Theia and sen2cor snow maps were compared …*", here Table 5 shows only the results related to Sentinel-2 dataset. Once again, please specify that the evaluations are based on Sentinel-2 data, as currently reported in Section 6 (see main remark n°5 in General comments).

Table 5: Please provide the definition of F1, FPR and FNR, how they have been evaluated and their optimal values (see main remark n°3 in General comments).

Pag. 19, l. 4-5: Please consider moving the definition of "*false positive*" and "*false negative*" pixels in Section 4.2.1, where these terms are introduced for the first time.

**Section 5 (Discussion)**

Pag. 21, l. 10: Please provide a reference, if available.

Pag. 21, l. 25: Please consider removing *of* preceding *the snow cover*.

Pag. 21, l. 26: According to my previous comment on Section 2, it would be useful to explain how the algorithm parameters have been defined.

**Section 6 (Conclusion)**

In this final section the authors provide useful information of key importance, which should be provided throughout the manuscript:

1. In the Introduction it should be explicitly specified that most snow products are derived from Sentinel-2 and only recently from an integration of Sentinel-2 and Landsat-8 data. As well, at the beginning of Section 4 the authors should explicitly state that the *evaluation was based only on Sentinel-2 products*.
2. In the section describing the methodology it should be properly explained that the analyses have been performed by considering the snow cover detected over the satellite field of view, and not at ground level (no correction of vegetation masking effect).
3. The condition of clear-sky should be explicitly mentioned as one criteria for the selection of the scenes to be analysed.

---

## Referee Comment (RC2) · Herrero (Referee) · 10 Jan 2019

**General comments**

The authors present a high quality snow product derived from satellite information that is of good use to a multitude of studies and applications related to snow. The fact that it is especially oriented to mountainous areas, where it is more difficult for this type of products to provide a high quality standard, is a good indicator of the validity of this product and the algorithms used.

The improvements introduced in the snow detection algorithm and the availability and

characteristics of the snow product seem very relevant and worthy of publication.

The manuscript is well and clearly written, with a careful editing, ordering and presentation of the concepts.

**Specific comments**

My main specific comment refers to the presentation of the Theia collection as a mixed product of Sentinel–2 and Landsat–8. It is true that it will be in the future, but currently, and in what is presented in this manuscript, for all intents and purposes, it is only Sentinel–2. Therefore, I would consider withdrawing references to Landsat–8, especially from the title.

Another comment arises on the fact that we do not understand that the validation with SPOT imagery was done for cloud-free data until page 16 (line 3). It is an important detail, which perhaps is not given due importance previously, because one of the presented strengths of the new algorithm is the treatment of the clouds. The validation for clouds discrimination lays on the visual verification, which is not as consistent as other methods. May be this can be addressed in future works, but it would be interesting at least that this issue became clearer throughout the manuscript.

The download of the Snow product through Theia portal (theia.cnes.fr) is a little bit messy in regard to the selection of tiles and dates (compared, for example, to Sentinel Hub portal). Nevertheless, I was able to download a particular image with a delay of only 1 day, and the accuracy of the snow mask was impressive (for a clear day with patchy snow over Sierra Nevada). Very few shaded or partially-covered snow pixels are lost (false negatives), while no false positives are apparently seen. The product is really high quality.

And a suggestion referred to the hard classification as snow/no snow pixels: obviously, the current resolution of Sentinel is superb. But maybe it would be worthwhile to generate some subpixel information, i.e. *fractional snow cover*. Probably with current

technology it is possible to generate this product that could be important in semi–arid areas where patchy snow prevails over large areas during snowmelt periods.

**Technical corrections**

- Page 4. Line 11. $n_i$ and $r_i$ are actually four parameters, not two.

- Page 4. Line 20. $f_t$ is introduced without any definition or reference to Table 1. Perhaps a reference to the table would be pertinent in this paragraph, where most of the parameters that appear in it are presented.

- Page 7. Figure 3. Caption. Where is Table 2.6?

- Page 8. Table 1. r1 appears twice. The last one should be r2, I guess.

- Page 16. Table 3. Caption. Why the reference to Fig. 3?

- Page 16. Table 3. What is the criterion followed to order the rows?

- Page 18. Line 22. Where is Fig. B4?
* * *

---

## Referee Comment (RC3) · Herrero (Referee) · 11 Jan 2019

I want to add two technical corrections to the previous post, regarding the format of the data downloaded from the Theia portal:

- Page 13. Line 6. It seems that the *productID_EXS_R2.tif* file is not provided in the zipped file right now. The zip file contains then 8 files in total.

- Page 13. Line 14. I think that you meant *a quicklook picture of the productID_**SNW**_R2.tif image*

---

## Referee Comment (RC4) · Dozier (Referee) · 14 Jan 2019

This paper appears to be quite intriguing. However, in order to review it, I want to access some of the results, but I am not able to create an account at https://theia.cnes.fr/. At this point, my comments are rather sparse, but I would be willing to review the paper again if this problem with creating an account can be rectified.

For the world's mountains, the "cirrus" band — Landsat 8 OLI (band 9), MODIS (band 26), or Sentinel-2 (band 10) — produces problematic output. The idea is that the wavelengths in this band are in a region of strong absorption by atmospheric water vapor, so that anything with high reflectance is likely a cirrus cloud. However, much of

the snow of interest is at high elevation! In some areas of the world, for example, the Himalaya, almost all the snow is misclassified as cirrus in the Landsat 8 BQA file.

One of the other referees has stated that the Theia archive mostly consistes of Sentinel-2A/B products, with little representation of Landsat 8 data. This is a shame, as the combination of the two systems provides imagery with a high temporal resolution.

In this context, there is no discussion of the CFMask for Landsat 8. References to consider are: Foga, S., Scaramuzza, P. L., Guo, S., Zhu, Z., Dilley Jr, R. D., Beckmann, T., Schmidt, G. L., Dwyer, J. L., Hughes, M. J., and Laue, B.: Cloud detection algorithm comparison and validation for operational Landsat data products, Remote Sensing of Environment, 194, 379-390, doi 10.1016/j.rse.2017.03.026, 2017. Zhu, Z., Wang, S., and Woodcock, C. E.: Improvement and expansion of the Fmask algorithm: cloud, cloud shadow, and snow detection for Landsats 4-7, 8, and Sentinel 2 images, Remote Sensing of Environment, 159, 269-277, doi 10.1016/j.rse.2014.12.014, 2015.

In our experience, some clouds have Landsat 8 OLI signatures that can be similar to some snow, for example thin cirrus clouds compared to fractional snow. Moreover, Selkowitz et al. have shown that in mountainous regions, the majority of Landsat snow pixels are not fully snow covered at 30 m resolution. Reference to consider is: Selkowitz, D. J., Forster, R. R., and Caldwell, M. K.: Prevalence of pure versus mixed snow cover pixels across spatial resolutions in alpine environments, Remote Sensing, 6, 12478-12508, doi 10.3390/rs61212478, 2014.

As mentioned previously, I can take a closer look when I can get acess to Theia.

---

## Short Comment (SC1) · 14 Jan 2019

Dear Dr Dozier,

thank you for these first comments. I am surprised that you were not able to create a Theia account. It happened in the past that the authentication server was not responding for a few hours. Can you try again now? You can always contact the support (exploitation.theia-land@cnes.fr) or get in touch with me if you experience any difficulty.

Best regards,

Simon

---

## Referee Comment (RC5) · Anonymous Referee #4 · 22 Jan 2019

Review of Gascoin et al.: Theia Snow collection: high resolution operational snow cover maps from Sentinel-2 and Landsat-8 data

General comments:
The authors present the algorithm for the derivation of high-resolution operational snow cover maps from Sentinel-2 images. They further present an evaluation using station measurements, fine-scale satellite-derived snow maps as well as visual control. Overall, they found good performances though some tendencies were revealed.
On cloudless days, the Theia snow product more likely underestimated snow coverage when compared to station measurements and fine-scale satellite-derived snow maps. When including cloudy days, the visual control with the false color composites revealed that the Theia snow product more likely overestimated snow coverage due to confusion with cloud coverage.

This manuscript presents a complete package consisting of the algorithm to derive a satellite-derived snow product, a thorough evaluation as well as the access point to freely download the snow products via the Theia portal. It is therefore interesting for a broad range of scientists and practitioners.
I already have an account at Theia and previously downloaded snow cover maps via the command line tools on Linux provided on github (https://github.com/olivierhagolle/theia_download).

The manuscript is well written and I have only minor issues, which should be addressed before publication.

Specific comments:

Since Landsat-8 data was not part of the snow product evaluation I would also suggest removing it everywhere (also in the title). However, I would keep mentioning it in the conclusions such that the reader is aware of the Landsat-8 snow maps.

It is only in the conclusion that the user learns he has to apply a forest mask to exclude forested areas from Theia snow maps. I think this should be mentioned earlier, e.g. as a bullet in the 2.1 Scope?

In section 2.2 you mention the correction for atmospheric and terrain slope effects in the Sentinel images. Can you cite something, since it is not explained or referenced later on?

In section 4.1.2 the authors present the evaluation with Spot images having a higher spatial resolution than Sentinel. I might not have fully understood the description of the Spot evaluation with Sentinel data. Did the size of each of the fifteen Spot polygons match one Sentinel pixel, such that fifteen pixels were evaluated for each of the six dates (or per tile)? If this is correct, how many of each were homogeneously snow-covered respectively snow free?

Two of the selected six dates were in August and in October, i.e. in late summer/early fall. How much snow cover was there on 2016-08-13 in tile

31TGK and on 2016-10-12 in tile 32TLS? I guess the snow coverage could have an impact on the performances?

Technical comments:

1) Please check all figure and table numbers, the references to them in the text as well as the order when they are introduced.

2) Page 19, line 5: Should this be: "False negative pixels (i.e. pixels falsely classified as snow-**free**) …" ?

3) Page 21, line 1: "..a**n** excellent accuracy.."

4) Page 21, line 6: "In addition, in situ measurements may not **be** representative of the snow conditions…"

---

## Author Comment (AC1) · 15 Mar 2019

The response letter and a revised version of the manuscript are provided in the attached zip file.

Please also note the supplement to this comment:
https://www.earth-syst-sci-data-discuss.net/essd-2018-144/essd-2018-144-AC1-supplement.zip

---

## Author Response (AR1)

**Response to referees [essd-2018-144]**

We would like to thank the four referees for their positive and relevant comments, which helped us improve the manuscript. We have followed most of the suggestions as explained below. All changes were marked in bold in the annotated version of the manuscript.

Simon Gascoin on behalf of all coauthors.

**Response to Gaia Piazzi (referee 1)**

GENERAL COMMENTS
The authors introduce a freely-available collection of snow products derived from Sentinel-2 and Landsat-8 data through a robust retrieval algorithm, which allows overcoming well-known limitations affecting the snow detection. The newly-developed algorithm is described in detail and its potential is tested and discussed through several analyses. The satellite-based dataset presented in this study is of critical importance and great usefulness not only for the scientific research community, but also for users interested in operational applications. I have identified some issues which deserve only minor revisions of the manuscript before its publication.

We appreciate this positive feedback and all the relevant suggestions below.

SPECIFIC COMMENTS
Here are listed my main comments:
1. The manuscript contains several self-explanatory figures, which make it easier to read and
understand the text. However, further references to figures need to be introduced, since for some of them the reference is even missing.
This was corrected (see specific comments below).

2. The manuscript comprehensively describes the newly-developed algorithm and the accuracy analysis of the resulting satellite products. However, a description of the satellite datasets used in this study (i.e., Sentinel-2 and Landsat-8) is missing. It would be of key interest to introduce the spectral bands of these satellite imagery to support the description of LIS algorithm.
We have added Tab. 2 to indicate the spatial resolution and central wavelength of the spectral bands used by LIS for each compatible sensor.

3. A further section introducing the evaluation metrics should be added to better enable the assessment of the results.
We have added Sect. 4.1.3 to describe the evaluation metrics.

4. Although the title mentions both Sentinel-2 and Landsat-8 data, references to Landsat-8 dataset are generally missing in the manuscript. Indeed, both methodology and results are actually referred to Sentinel-2 imagery.

todo

5. The final section 6 includes information of critical importance, which should be reported and
discussed in the text. Please consider my comments on Section 6.
We have responded in the specific comment on Sect. 6 below.

Please consider replacing sen2cor with Sen2Cor throughout the whole text.
Done
Section 1 (Introduction)
Pag. 1, l. 20: Please report the reference within brackets.
Done
Pag. 1, l. 22: Please replace 12h with 12 hours.
Done
Pag. 2, l. 5: Please consider reporting the full name of the plan.
The full reference is given in the reference list
Pag. 2, l. 10: Please report the reference within brackets.
Done
Pag. 2, l. 14: Please consider replacing 5 day with 5-days.
Done (*"of 5 days"*)
Pag. 2, l. 19, 21: Please explicitly define and explain what level-1C and level-2A products are (i.e., TOA, BOA).
Done
*"(...) level-1C tiled product (a monodate ortho-rectified image expressed in top-of-atmosphere reflectance)*
*(...) level 2A (L2A) products (monodate ortho-rectified images expressed in surface reflectance, including a cloud mask)"*
Pag. 2, l. 21: Please specify the starting sensing date of the processed L2A product provided by ESA.
It was written "Since March 2018 ..."
Pag. 2, l. 24: What are the main limitations of ESA's L2A algorithm for snow detection?
As explained below, we prefer to keep this for the Discussion. We have added the following paragraph in Sect. 5:
*"Both Sen2Cor and LIS use the NDSI to detect the snow cover, however the algorithms differ in many aspects. The better performances of LIS in Sect. 4.2.2 can be related to the mono-date approach for atmospheric correction and cloud detection in Sen2Cor, which can cause snow/cloud confusion in the L2A product, while LIS uses L2A products from the more accurate multi-temporal MAJA processor (Baetens et al. 2019). In addition, a unique feature of the LIS algorithm is the use of the snow line elevation concept to improve the robustness of the snow detection in mountain regions. In addition, LIS was designed to retrieve snow below thin clouds and to reclassify snow pixels which are frequently found near the snow cover edges in L2A products. Apart from the MAJA processing, however, the effect of theses algorithmic differences could be less evident in flat areas."*

Pag. 2, l. 28: Please consider explicitly specifying here that Theia collection relies on a combination of these two satellite datasets. Currently this information is more clearly reported only in the Conclusions (see main remark n°5 in General comments).

We agree that this was not clearly presented. We have added the specific comment in the introduction:

*"The Theia collection was recently upgraded to also provide snow cover maps at 30 m resolution from Landsat-8 data using the same algorithm."*

Pag. 2, l. 28: Please consider rephrasing the sentence as "... at 20-m and 30-m resolution, respectively...".

The sentence was split in two parts to clarify.

Pag. 2, l. 30: Please report here the reference for the Normalized Difference Snow Index.

Done

Pag. 2, l. 31: Please close the bracket.

Done

Section 2 (Algorithm)

Please add in the text references to Figures 1 and 2, which need to be commented (see main remark n°1 in General comments).

We have added a reference to Fig. 1 and 2 in Section 2.5:

*"An example of the evolution of the snow line elevation over the tile 31TCH in the Pyrenees is shown in Fig. 1 and the corresponding snow maps are shown in Fig. 2."*

Please refer to Table 1 whenever an algorithm parameter is introduced.

We have added several references to Tab. 1. We also added a comment in the end of Sect. 2.4.1: *"The values of the parameters  are provided in Tab. 1."* Note that the text often refers to Fig. 3 (flow chart), whose caption also refers to Tab. 1 (parameters).

Generally, in Section 2 the authors do not explain how the parameters values have been defined and optimized. Please add this missing information in the algorithm description.

We added the following sentence:

*"The parameter values were set based on previous studies with Landsat (Hagolle et al., 2010; Zhu et al., 2015; Gascoin et al., 2015) and by visually checking many snow maps and snow fraction histograms. From this set of a priori parameter values, only r2 was adjusted based on the analysis of a first batch of products to enhance the snow detection in shaded slopes."*

Section 2.1 (Scope)

Currently some limitations of the ESA's processor are discussed within the description of LIS algorithm, for instance the critical issues affecting the cloud mask (i.e., thin clouds, edges of snow cover). In order to improve the readability of the manuscript and to properly highlight the potential of LIS algorithm, the authors could clearly identify and explain in this section what the main limitations of the ESA's processor are, before describing their algorithm.

We have started to design LIS before Sen2Cor was available. We agree that it is important to discuss the algorithmic differences between LIS and Sen2Cor to understand the differences in the results, however we prefer to do it as part of the Discussion (a paragraph was added).

Section 2.2 (Input)

In this section the input data are listed, but the data processing to derive input L2A product is not discussed. This section should also briefly explain how the correction of atmospheric and terrain slope effects is performed.

For the sake of simplicity we prefer to refer to MAJA ATBD where all the details are given (reference added in Sect. 2.2).

Pag. 3, l. 13: Please consider replacing remote sensing images with remotely-sensed images.

Done (remotely sensed)

Section 2.3 (Pre-processing)

Pag. 3, l. 22: Please specify the original spatial resolution of the red and green bands. It would be useful to introduce a table reporting the spectral bands and their central wavelengths of the original satellite data (see main remark n°2 in General comments).

Done (new Tab. 2)

Pag. 3, l. 23: Please spell out SRTM.

Done

Section 2.4.1 (Snow detection)

Are the cloud-contaminated pixels preliminary flagged and neglected in this first procedure? If so, please specify this information.

We have indicated "*a cloud-free pixel*"

Section 2.5 (Snowline elevation)

Please consider renumbering this Section as 2.4.2.

Done. Thank you, this was a mistake.

Pag. 4, l. 15: How are the false snow detections defined? What is the reference dataset to identify false snow detections?

At this stage there is no reference data, it is a guideline we adopt to have as much as possible true snow pixels.

Pag. 4, l. 18: Sentinel-2 tile is 100 km 2 .

No it is 110 km by 110 km (109.8 km), about 12E3 km².

Pag. 4, l. 20: Please specify how snow fraction is evaluated (fraction evaluated over the total number of snow and no-snow pixels?).

We have added "*the total snow fraction in the image*".

Furthermore, how is defined the threshold of snow fraction f t ?

Pag. 4, l. 23-24: How are defined d z and f s ?

We have added references to Tab. 1.

f_t = 0.1% was defined based on the fact that it corresponds to 30'000 pixels, which we considered sufficient to have a statistically robust estimation of the snow line in the image. dz=100 m and f_s=10% are indicated in Tab. 1. These parameters were defined in the early development phase by visually checking the snow maps and the snow fraction histogram (see response to a similar comment above). Here we have separated in the text the algorithm description to the parameters values because LIS is easily configurable to change these parameters. We did not test the sensitivity of these parameters using reference data but this could be the focus of further work.

Section 2.6 (Cloud mask processing)

Please consider renumbering this Section as 2.4.3.

Done

Pag. 5, l. 15: Please either specify how factor r f is defined or neglect this information by just reporting the corresponding reference.

We prefer to keep iti because r_f is a parameter like the others (it must be defined in the LIS configuration file), and we need to introduce it in the algorithm description.

Pag. 6, l. 2, 6: Please specify how r D and r B are defined.

We have added references to Tab. 1.

Figure 3: Please introduce MUSCATE in text before this figure.

We have added this sentence in the Figure caption: "*MUSCATE is the scheduler which manages the L2A production for Theia.*"

Section 2.7 (Implementation)
Please consider renumbering this Section as 2.4.4.

Done

Section 3 (Data description)
Pag. 9, l. 3: The number of selected tiles is already reported in pag. 8, l. 15.

We would rather let it here.

Pag. 12, l. 12: Please specify how the pixels classified as no-snow are defined. Are all snow- and cloud-free pixels classified as no-snow ones? Does the no-snow class include water pixels? Here it should be also mentioned the uncertainty in classifying vegetated pixels as snow or no-snow ones.

Indeed no-snow pixels can be anything except no-data, cloud or snow. Regarding the uncertainties in forest regions we do not think that it should mentioned here since this section is devoted to the data description. We have highlighted this aspect in Sect. 6.

Pag. 12, l. 13: Please specify what pixels are classified as no-data.

These are the pixels outside of the acquisition segment.

We have added : "*Note that no-snow pixels can be any surface, including water surface. No data pixels are the pixels outside of the acquisition segment.*"

Please switch Figure 5 and 6 since they are mentioned reversely in the text.

Actually Fig. 5 is mentioned first, but in the caption of the Fig.1 and 2.

Section 4.1 (Method)
Since in this section the authors refer only to Sentinel-2 (e.g., pag. 13, l. 28; Table 3), they should explicitly specify that the evaluation is based only on Sentinel-2 data, as clearly stated in Section 6 (see main remark n°5 in General comments).

This is a good suggestion. We have moved this sentence from the conclusion to the beginning of the Method section: "*The evaluation was based only on Sentinel-2 products, because the production for Landsat-8 had just started at the time of writing and thus the Landsat-8 products represented a very minor fraction of the Theia Snow collection.*"

As this manuscript is focused on the mapping of snow cover, it would be more proper to limit the analysis over the snow period, not up to the end of summer season. For instance, Table 3 reports a selected pair of snow products in August, which is not of high interest for the main purpose of this research. Furthermore, it is noteworthy that, when assessing the contingency matrix, the analysis of these snow-free scenes could affect the representativeness of the results, as shown by the significantly higher number of correct negatives in Table 4.

We think it is important to assess the snow product in different seasons. In this product we find 174603 snow covered pixels mainly in the Parc National des Ecrins. It is only 0.6% of the image area, but still 69.8 km² (see Fig. R1 below). Regarding the representativeness of the results we agree that the accuracy may not be an adequate performance measure when the number of no-snow pixels is much greater than the number of snow pixels, however we have also computed the kappa coefficient and F1-score to account for this.

[Figure]

Fig R1: zoom over the Ecrins and Oisans on 2016-Aug-13 (snow covered outline: magenta, clouds: green).

Section 4.1.1 (Comparison with in situ snow depth measurements)
Please add in the text references to Figures 7 and 8, which need to be commented (see main remark n°1 in General comments).
We thank the referee for noting this. Now Fig. 7 is referred to in Sect. 3 and Fig. 8 in Sect. 4.1.1.
Pag. 15, l. 2: How is defined the first tile? Are the Sentinel-2 data of the second tile neglected even when sensed in a different day with respect to the selected first tile? If so, it would entail a loss of useful remotely-sensed snow data.
Given the overlap of between Sentinel-2 tiles (Fig. 8), it can happen that a station is located in two adjacent tiles. We did not assess the differences between selecting one or another tile, however we do not expect significant differences since the Sentinel-2 L1C data are the same. By visual inspection we did not detect discontinuities in the snow cover between tiles, which suggests that the LIS algorithm is robust to the tiling scheme. We aim to study this more extensively in the future by comparing the snow pixels in the overlapping areas independently of the station dataset.
Section 4.1.2 (Comparison with snow maps of higher spatial resolution)
Pag. 16, l. 11: Please remove the.
Done
3Pag. 16, l. 11: Could the authors mention the classification algorithms that have been tested and provide their references?
We tested different algorithms in the Orfeo Toolbox: k-means, decision tree, random forest, support vector machine and gradient boosted trees (boost). We do not think that it is useful to specify this given that we use only the results of the boost method here. However the interested reader can check the details of this work in the master thesis report of Bouchet (2017) that is cited in the manuscript (and available online).
Section 4.2.1 (Comparison with in situ snow depth measurements)

To improve the assessment of the results, please report how the evaluation metrics are defined (see main remark n°3 in General comments).

We have added a new section 4.1.3 for this purpose.

Section 4.2.2 (Comparison with snow maps of higher spatial resolution)

While in Section 4.1.2 the authors explain that "The Theia and sen2cor snow maps were compared ...", here Table 5 shows only the results related to Sentinel-2 dataset. Once again, please specify that the evaluations are based on Sentinel-2 data, as currently reported in Section 6 (see main remark n°5 in General comments).

This is now clearly specified at the beginning of Sect. 4.

Table 5: Please provide the definition of F1, FPR and FNR, how they have been evaluated and their optimal values (see main remark n°3 in General comments).

Done (Sect 4.1.3).

Pag. 19, l. 4-5: Please consider moving the definition of "false positive" and "false negative" pixels in Section 4.2.1, where these terms are introduced for the first time.

We believe that it could help the reader to interpret the results to let it here as well.

Section 5 (Discussion)

Pag. 21, l. 10: Please provide a reference, if available.

Done (Gascoin et al. 2015)

Pag. 21, l. 25: Please consider removing of preceding the snow cover.

Done

Pag. 21, l. 26: According to my previous comment on Section 2, it would be useful to explain how the algorithm parameters have been defined.

This is now explained in Sect. 2.4.4 (see response to similar comments above).

Section 6 (Conclusion)

In this final section the authors provide useful information of key importance, which should be provided throughout the manuscript:

1. In the Introduction it should be explicitly specified that most snow products are derived from
Sentinel-2 and only recently from an integration of Sentinel-2 and Landsat-8 data.

Done

As well, at the beginning of Section 4 the authors should explicitly state that the evaluation was based only on Sentinel-2 products.

Done

2. In the section describing the methodology it should be properly explained that the analyses have been performed by considering the snow cover detected over the satellite field of view, and not at ground level (no correction of vegetation masking effect).

This is an important comment and we are thinking of adjusting the snow maps based on the tree cover density for future products. However since this journal focuses on the presentation of data, we deliberately gathered all the limitations of the product in the conclusion to provide a synthetic caveat to the user.

3. The condition of clear-sky should be explicitly mentioned as one criteria for the selection of the scenes to be analysed.

If the referee refers to the Theia vs. SPOT snow maps, this was specified in Sect. 4.1.2 "*We searched in the catalogue of the Kalideos database for available SPOT images that could match a cloud-free (or nearly cloud-free) Theia snow map*".

**Response to Javier Herrero (referee 2)**

General comments
The authors present a high quality snow product derived from satellite information that is of good use to a multitude of studies and applications related to snow. The fact that it is especially oriented to mountainous areas, where it is more difficult for this type of products to provide a high quality standard, is a good indicator of the validity of this product and the algorithms used.

The improvements introduced in the snow detection algorithm and the availability and characteristics of the snow product seem very relevant and worthy of publication. The manuscript is well and clearly written, with a careful editing, ordering and presentation of the concepts.

We are grateful to the referee for this positive appreciation of our work.

Specific comments
My main specific comment refers to the presentation of the Theia collection as a mixed product of Sentinel–2 and Landsat–8. It is true that it will be in the future, but currently, and in what is presented in this manuscript, for all intents and purposes, it is only Sentinel–2. Therefore, I would consider withdrawing references to Landsat–8, especially from the title.

We agree that our manuscript largely focuses on Sentinel-2. That is the main novelty of our work and indeed our main motivation. In addition, the operational production began with Sentinel-2 products, therefore most of the available products at the time of writing were generated from Sentinel-2. However, we have developed the LIS processor to be compatible with Landsat-8 and an increasing number of Landsat-8 products are being made available in Theia. Therefore we would like to keep Landsat-8 in the manuscript. We have modified the abstract ("*We present the algorithm to generate the snow products and provide an evaluation **of Sentinel-2 snow products** accuracy*") and conclusion to clarify this point:

*The Theia Snow collection is a free collection of snow products which indicate the presence or absence of snow on the land based on Sentinel-2 (20~m) and Landsat-8 (30~m) data. Most of the snow products are derived from Sentinel-2. **However, in late September 2018, Landsat-8 snow products started being routinely added to the Theia Snow collection. Previous Landsat-8 data were also reprocessed back to March 2017. At the time of writing (15 March 2019), for example in tile T31TCH in the Pyrenees, 20.5% of all available products were derived from Landsat-8 data (63 among 244). In addition, the processing is done to facilitate the combination of Landsat-8 and Sentinel-2 snow products since the data format and the tiling scheme are the same (only the pixel size differs). Although the co-registration of Sentinel-2 and Landsat-8 data can be still problematic (Storey et al. 2017), the combination of "analysis ready" Landsat-8 and Sentinel-2 is important to increase the number of cloud-free observations.***

(...) it remains a preliminary assessment that should be extended in the future. For example, the evaluation was based only on Sentinel-2 products, **since the integration of Landsat-8 data started during the writing of this manuscript and should be extended to Landsat-8 products. However, the processing is done with the same algorithm, which is based on the calculation of the NDSI from slope-corrected surface reflectance (level 2A data).**

Another comment arises on the fact that we do not understand that the validation with SPOT imagery was done for cloud-free data until page 16 (line 3). It is an important detail, which perhaps is not given due importance previously, because one of the presented strengths of the new algorithm is the treatment of the clouds. The validation for clouds discrimination lays on the visual verification, which is not as consistent as other methods. May be this can be addressed in future works, but it would be interesting at least that this issue became clearer throughout the manuscript.

Unfortunately very-high resolution images are generally not acquired when the sky is cloudy to save production cost. We agree that this is an important limitation of the evaluation and we have specified "very-high resolution **clear-sky** snow maps" in the introduction of the new version of manuscript. Note however that there are still some clouds in the SPOT scenes (but a very minor fraction of the image). There are some clouds too in the Sentinel-2 products, which were correctly identified (e.g. green polygons in Fig R1 below). Note that in the meantime, the MAJA cloud mask has been thoroughly validated using an active learning approach (Baetens et al. 2019), showing the high accuracy of MAJA cloud mask, which outperforms the one from Sen2Cor. This was added in Sect. 5.

[Figure]

Fig R1: zoom over the Ecrins and Oisans on 2016-Aug-13 (snow covered outline: magenta, clouds: green).

The download of the Snow product through Theia portal (theia.cnes.fr) is a little bit messy in regard to the selection of tiles and dates (compared, for example, to Sentinel Hub portal).

Nevertheless, I was able to download a particular image with a delay of only 1 day, and the accuracy of the snow mask was impressive (for a clear day with patchy snow over Sierra Nevada). Very few shaded or partially-covered snow pixels are lost (false negatives), while no false positives are apparently seen. The product is really high quality.

We thank the referee for this feedback.

And a suggestion referred to the hard classification as snow/no snow pixels: obviously, the current resolution of Sentinel is superb. But maybe it would be worthwhile to generate some subpixel information, i.e. fractional snow cover. Probably with current technology it is possible to generate this product that could be important in semi–arid areas where patchy snow prevails over large areas during snowmelt periods.

We take note of this suggestion, which echoes a recent call by the European Environment Agency to produce near-real time Fractional Snow Cover (FSC) at 20 m based on Sentinel-2 NDSI. This is indeed a perspective of our work since it would be straightforward to implement a simple relationship FSC=f(NDSI) in LIS in order to provide the fractional snow cover area instead of a binary value (snow/no-snow). However, there is no current consensus on the shape and stability of such a relationship at the resolution of Sentinel-2 and further work shall be conducted to establish a robust algorithm.

Technical corrections
• Page 4. Line 11. n i and r i are actually four parameters, not two.
This is right, we changed to "parameter pairs".
• Page 4. Line 20. f t is introduced without any definition or reference to Table 1. Perhaps a reference to the table would be pertinent in this paragraph, where most of the parameters that appear in it are presented.
We have added references to the parameter Table 1 every time that a new parameter is introduced. This was also suggested by Referee 1.
• Page 7. Figure 3. Caption. Where is Table 2.6?
Corrected, it was Tab 1.
• Page 8. Table 1. r1 appears twice. The last one should be r2, I guess.
Corrected, thank you.
• Page 16. Table 3. Caption. Why the reference to Fig. 3?
Corrected, it was Fig 9...
• Page 16. Table 3. What is the criterion followed to order the rows?
We sorted the rows by date in this revision.
• Page 18. Line 22. Where is Fig. B4?
It was Fig. A1 in Appendix. We apologize for all these confusions in the Figure labels.
I want to add two technical corrections to the previous post, regarding the format of the data downloaded from the Theia portal:
• Page 13. Line 6. It seems that the productID_EXS_R2.tif file is not provided in the zipped file right now. The zip file contains then 8 files in total.
Thank you for checking. Actually it depends on the product version. This file was stored in the root of the zip file in v 1.0, but in more recent versions it is placed in the MASKS folder

within the zip file. We have modified the text and also added this item which was missing in the product description:

"- `DATA/productID_HIS_R2.txt` *(since product version 1.4): a text file indicating the cloud, snow and no-snow fraction area by elevation bins (Sect. 2)."*

See the product version details here: ([http://www.cesbio.ups-tlse.fr/multitemp/?page_id=13180](http://www.cesbio.ups-tlse.fr/multitemp/?page_id=13180)).

• Page 13. Line 14. I think that you meant a quicklook picture of the productID_SNW_R2.tif image

We corrected this mistake, thank you!

**Response to Jeff Dozier (referee 3)**

This paper appears to be quite intriguing. However, in order to review it, I want to access some of the results, but I am not able to create an account at https://theia.cnes.fr/. At this point, my comments are rather sparse, but I would be willing to review the paper again if this problem with creating an account can be rectified.

We are sorry that the referee could not rectify this problem during the review process.

For the world's mountains, the "cirrus" band — Landsat 8 OLI (band 9), MODIS (band 26), or Sentinel-2 (band 10) — produces problematic output. The idea is that the wavelengths in this band are in a region of strong absorption by atmospheric water vapor, so that anything with high reflectance is likely a cirrus cloud. However, much of the snow of interest is at high elevation! In some areas of the world, for example, the Himalaya, almost all the snow is misclassified as cirrus in the Landsat 8 BQA file.

The cirrus detection with the 1.38 µm band is handled by the MAJA processor. The detection is done using a threshold which increases with altitude (see Hagolle et al., 2017 MAJA Algorithm Theoretical Basis Document http://doi.org/10.5281/zenodo.1209633)

One of the other referees has stated that the Theia archive mostly consisted of Sentinel-2A/B products, with little representation of Landsat 8 data. This is a shame, as the combination of the two systems provides imagery with a high temporal resolution.

The integration of Landsat-8 began in October 2018 and therefore the number of Landsat-8 products was not significant during the preparation of this manuscript. Since then the Landsat-8 data have been routinely produced and reprocessed back to March 2017. Therefore, we have modified the manuscript to highlight that a significant fraction (about 20% depending on the tile) of the products in the Theia Snow collection now consists of Landsat-8 products.

In this context, there is no discussion of the CFMask for Landsat 8. References to consider are: Foga, S., Scaramuzza, P. L., Guo, S., Zhu, Z., Dilley Jr, R. D., Beckmann,T., Schmidt, G. L., Dwyer, J. L., Hughes, M. J., and Laue, B.: Cloud detection algorithm comparison and validation for operational Landsat data products, Remote Sensing of Environment, 194, 379-390, doi 10.1016/j.rse.2017.03.026, 2017. Zhu, Z., Wang, S.,and Woodcock, C. E.: Improvement and expansion of the Fmask algorithm: cloud,cloud shadow, and snow detection for Landsats 4-7, 8, and Sentinel 2 images, Remote Sensing of Environment, 159, 269-277, doi 10.1016/j.rse.2014.12.014, 2015. In our experience, some clouds have Landsat 8 OLI signatures that can be similar to some snow, for example thin cirrus clouds compared to fractional snow.

We checked outputs of the Fmask algorithm (Zhu et al. 2015) during the early phase of the project and we also found issues in mountain regions in particular confusions between shaded slopes and water surface, snow and cloud confusion (Fig. R2 below). This also motivated us to develop our own algorithm using slope-corrected surface reflectances.

[Figure]

Figure R2: comparison of LIS and Fmask outputs for a Landsat-8 scene in the Central Pyrenees (2014-Jan-11).

Moreover, Selkowitz et al. have shown that in mountainous regions, the majority of Landsat snow pixels are not fully snow covered at 30 m resolution. Reference to consider is:Selkowitz, D. J., Forster, R. R., and Caldwell, M. K.: Prevalence of pure versus mixed snow cover pixels across spatial resolutions in alpine environments, Remote Sensing,6, 12478-12508, doi 10.3390/rs61212478, 2014.

This is a very interesting study and we thank the referee for bringing it to our attention. The results suggest that in the study area with maritime climate the fraction of mixel pixels at the Sentinel-2 resolution is below 40%, whereas it can be 80% in a more continental mountain area. We also started looking at this issue recently by analyzing the fractional snow cover area from 6 m resolution SPOT-6 snow maps in the French Alps. The SPOT-6 sensor has a lower resolution than the one of WorldView used by Selkowitz et al. (2014) but covers larger areas (swath width at nadir: 60 km vs. 14.6 km). Our results (Fig. R4 below) suggest that the fraction of mixed pixels in lower. This might be due to the lower resolution of the original snow map (6 m) and the fact that Selkowitz et al. (2014) defined mixed pixels as pixels with FSC between 0.02 and 0.98, while in Fig. R4 we adopted a less strict representation by

choosing FSC bins of 10%. However, current state-of-the-art FSC algorithm do not reach an accuracy as low as 0.02. In another study, Selkowitz et al. 2017 report a root mean square error in FSC of 0.2. ("The USGS Landsat Snow Covered Area Products: Methods and Preliminary Validation." Available online). Therefore the mixel pixel definition in the pioneer study by Selkowitz et al. (2014) seems rather strict for an assessment of the error that would be done in operational context by using binary snow maps instead of FSC.

[Figure]

Figure R3: Fraction of partially snow-covered pixels in snow covered pixels, for pixel resolutions between 1 m and 500 m, for (**top**) a study area in the Oregon Cascades, and (**bottom**) a study area in the Rocky Mountain NP. Vertical red line indicates 20 m spatial resolution (Sentinel-2 Theia snow products). Vertical blue lines indicate 40 m spatial resolution (slightly above the nominal spatial resolution for Landsat) and 500 m spatial resolution (the nominal spatial resolution for MODIS) (source: modified from Selkowitz et al. *Remote Sens.* **2014**, *6*(12), 12478-12508; doi:10.3390/rs61212478)

[Figure]

Figure R4: Fraction snow cover (FSC in %) from a SPOT-6 image at 6 m resolution in the French Alps (Oisans). The SPOT-6 FSC was computed a follows: (1) supervised classification of the image in snow and no-snow classes from the multispectral and panchromatic data at 1.5 m and 6 m resolutions (2) resampling of the binary snow cover map to a FSC map at the same resolution as Sentinel-2 SWIR band (20 m) by spatial averaging.

As we replied to Referee 2 above, we are actually considering to distribute near-real time FSC at 20 m based on Sentinel-2. It would be straightforward to implement a simple relationship FSC=f(NDSI) in the LIS code. However, there is no current consensus on the shape and stability of such a relationship at the resolution of Sentinel-2 and further work shall be conducted to evaluate if a robust algorithm can be obtained from this function or if a more computer-intensive approach based on spectral unmixing should be considered (within the constraints of operational production).

As mentioned previously, I can take a closer look when I can get access to Theia.

We are sorry that the referee could not access the data during the period of open discussion. However, we are grateful to the referee for his insightful comments and we remain available for further discussion or assistance in accessing the data even after the open discussion of this paper.

**Response to Referee 4**

General comments:
The authors present the algorithm for the derivation of high-resolution operational snow cover maps from Sentinel-2 images. They further present an evaluation using station measurements, fine-scale satellite-derived snow maps as well as visual control. Overall, they found good performances though some tendencies were revealed. On cloudless days, the Theia snow product more likely underestimated snow coverage when compared to station measurements and fine-scale satellite-derived snow maps. When including cloudy days, the visual control with the false color composites revealed that the Theia snow product more likely overestimated snow coverage due to confusion with cloud coverage. This manuscript presents a complete package consisting of the algorithm to derive a satellite-derived snow product, a thorough evaluation as well as the access point to freely download the snow products via the Theia portal. It is therefore interesting for a broad range of scientists and practitioners. I already have an account at Theia and previously downloaded snow cover maps via the command line tools on Linux provided on github (https://github.com/olivierhagolle/theia_download). The manuscript is well written and I have only minor issues, which should be addressed before publication.

We appreciate the positive comments and we are pleased that the referee downloaded some products. We also thank the referee for the suggestions below.

Specific comments:
Since Landsat-8 data was not part of the snow product evaluation I would also suggest removing it everywhere (also in the title). However, I would keep mentioning it in the conclusions such that the reader is aware of the Landsat-8 snow maps.

As commented to Referee 2 we would like to keep Landsat-8 in the title because these products now represent a significant fraction of the Theia Snow collection (about 20%). We have modified the abstract to make it clear that the evaluation was done only with Sentinel-2 products. The conclusion was also modified to highlight this point.

It is only in the conclusion that the user learns he has to apply a forest mask to exclude forested areas from Theia snow maps. I think this should be mentioned earlier, e.g. as a bullet in the 2.1 Scope?

We agree with the referee. We have added this in Section 2.1: "*An algorithm was designed to determine the snow presence or absence from Sentinel-2 and Landsat-8 observations outside areas of dense forest (...)*"

In section 2.2 you mention the correction for atmospheric and terrain slope effects in the Sentinel images. Can you cite something, since it is not explained or referenced later on?

We have added a reference to MAJA Algorithm Theoretical Basis Document.

In section 4.1.2 the authors present the evaluation with Spot images having a higher spatial resolution than Sentinel. I might not have fully understood the description of the Spot evaluation with Sentinel data. Did the size of each of the fifteen Spot polygons match one Sentinel pixel, such that fifteen pixels were evaluated for each of the six dates (or per tile)? If this is correct, how many of each were homogeneously snow-covered respectively snow free?

These polygons were not used for Sentinel-2. The fifteen polygons refer to the training data in the SPOT images that were used to perform a supervised classification of the snow cover area in the SPOT images. We hope that this clarifies this section.

Two of the selected six dates were in August and in October, i.e. in late summer/early fall. How much snow cover was there on 2016-08-13 in tile 131TGK and on 2016-10-12 in tile 32TLS? I guess the snow coverage could have an impact on the performances?

As responded to Referee 1, we find 174'603 snow covered pixels in the August product, mainly in the high elevation glacierized areas of Parc National des Ecrins, i.e. 0.6% of the image area (69.8 km²). In the October product we find 7.1% (856 km²) of the image pixels are classified as snow. We accounted for this asymmetry in the number of snow/no-snow by using the kappa coefficient and F1-score. Note that it is equally important to have a correct classification of snow and no-snow pixels. Therefore, an image with a small snow covered area has the same value as an image with a small snow-free area (e.g. winter image). With a summer image we validate that the snow product does not contain false snow detection (e.g. turbid lakes, bright urban areas, etc.)

Technical comments:
1) Please check all figure and table numbers, the references to them in the text as well as the order when they are introduced.
We apologize for the mistakes in the figures and table references. We have revised this thoroughly, also thanks to the comments of the other referees.
2) Page 19, line 5: Should this be: "False negative pixels (i.e. pixels falsely classified as snow-free) ..." ?
Thank you for noting this error, now corrected.
3) Page 21, line 1: "..an excellent accuracy.."
Corrected.

[revised manuscript text omitted]